# 5-aminosalicylic acid suppresses osteoarthritis through the OSCAR-PPARγ axis

Jihee Kim [1,2,8], Gina Ryu [1,8], Jeongmin Seo [1], Miyeon Go[1], Gyungmin Kim[1], Sol Yi[1], Suwon Kim[3], Hana Lee[4], June-Yong Lee[5], Han Sung Kim[4], Min-Chan Park[6], Dong Hae Shin[3], Hyunbo Shim [1], Wankyu Kim[1] & Soo Young Lee [1,2,7] ✉

Osteoarthritis (OA) is a progressive and irreversible degenerative joint disease that is characterized by cartilage destruction, osteophyte formation, subchondral bone remodeling, and synovitis. Despite affecting millions of patients, effective and safe disease-modifying osteoarthritis drugs are lacking. Here we reveal an unexpected role for the small molecule 5-aminosalicylic acid (5-ASA), which is used as an anti-inflammatory drug in ulcerative colitis. We show that 5-ASA competes with extracellular-matrix collagen-II to bind to osteoclast-associated receptor (OSCAR) on chondrocytes. Intra-articular 5-ASA injections ameliorate OA generated by surgery-induced medial-meniscus destabilization in male mice. Significantly, this effect is also observed when 5-ASA was administered well after OA onset. Moreover, mice with DMM-induced OA that are treated with 5-ASA at weeks 8–11 and sacrificed at week 12 have thicker cartilage than untreated mice that were sacrificed at week 8. Mechanistically, 5-ASA reverses OSCAR-mediated transcriptional repression of PPARγ in articular chondrocytes, thereby suppressing COX-2-related inflammation. It also improves chondrogenesis, strongly downregulates ECM catabolism, and promotes ECM anabolism. Our results suggest that 5-ASA could serve as a DMOAD.

Osteoarthritis (OA) is an age-related chronic degenerative joint disease that is characterized by cartilage destruction, osteophyte formation, subchondral-bone remodeling, and synovitis[1]. Although it is the most common degenerative joint disease globally and a leading cause of disability and reduced quality-of-life in older people[2], therapeutic options mainly involve pain management[3]. Thus, disease-modifying OA drugs (DMOADs) that both halt OA progression and restore joint structures/functions are urgently needed.

An OA hallmark is progressive destruction of cartilage extracellular matrix (ECM)[4]. Normally, this ECM is respectively synthesized and destroyed by chondrocyte-derived anabolic and catabolic factors in equilibrium, resulting in cartilage homeostasis. In OA, this balance is tipped towards catabolism mediated by the matrix-metallopeptidases MMP3, MMP9, MMP13, and the aggrecanase ADAMTS5[5,6]. These degrade collagen-II and aggrecan (ACAN), whose synthesis is also downregulated in OA[7,8]. Thus, targeting chondrocyte catabolic/

[1]Department of Life Science, Ewha Womans University, Seoul, Republic of Korea. [2]The Research Center for Cellular Homeostasis, Ewha Womans University, Seoul, Republic of Korea. [3]Department of Pharmacy, Ewha Womans University, Seoul, Republic of Korea. [4]Department of Biomedical Engineering, Yonsei University, Wonju, Republic of Korea. [5]Department of Microbiology and Immunology, Institute for Immunology and Immunological Diseases, and Brain Korea 21 PLUS Project for Medical Sciences, Yonsei University College of Medicine, Seoul, Republic of Korea. [6]Division of Rheumatology, Department of Internal Medicine, Yonsei University College of Medicine, Seoul, Republic of Korea. [7]Multitasking Macrophage Research Center, Ewha Womans University, Seoul, Republic of Korea. [8]These authors contributed equally: Jihee Kim, Gina Ryu. ✉e-mail: leesy@ewha.ac.kr

anabolic processes, thereby modifying cartilage-ECM structures, may yield promising OA therapies.

OSCAR is a receptor[9] that binds to the triple-helical peptide GPOGPAGFO of collagen-II[10] via its extracellular domains[11,12]. It is strongly expressed by osteoclasts, and its binding to collagen-II co-stimulates FcRγ, thereby upregulating signaling pathways that induce osteoclastogenesis[13]. Interestingly, we showed recently that although normal chondrocytes express OSCAR at negligible levels, they can be upregulated and may participate in OA: OA-cartilage chondrocytes express more OSCAR than normal in both mice and humans[14,15]. Moreover, as reported in the present study, simply injecting OSCAR-expressing adenovirus (Ad-OSCAR) into naive mouse joints induced OA-like disease. We, therefore hypothesized that small molecules that block OSCAR-collagen binding could be DMOAD candidates. This was tested in the present study by screening 3287 compounds.

One of these inhibited the collagen binding of OSCAR on chondrocytes, namely, 5-aminosalicylic acid (5-ASA; also known as mesalamine or mesalazine). 5-ASA is widely used to treat inflammatory bowel diseases (IBDs)[16], particularly ulcerative colitis (UC)[17]. We found that intra-articular (IA) 5-ASA injections significantly improved not only Ad-OSCAR-induced OA-like disease but also a classical murine model of OA, namely, surgery-induced medial-meniscus destabilization (DMM)[18]. Importantly, 5-ASA treatment also ameliorated DMM-induced OA when the injections started long after disease initiation. Late injections at weeks 8–11 also led to thicker cartilage compared to untreated mice that were sacrificed at week 8. Our RNA-seq, in vitro, and in vivo analyses showed that 5-ASA may improve OA via multiple molecular mechanisms. First, it upregulated PPARγ, which inhibited the pro-inflammatory eicosanoid pathway. Second, it enhanced the chondrogenic differentiation of mesenchymal stem cells (MSCs), which could promote cartilage regeneration. Third, it upregulated cartilage-specific ECM-anabolism and downregulated ECM-catabolism. Thus, OSCAR may play a key role in OA pathogenesis and 5-ASA may have significant potential as an IA-administered DMOAD.

## Results

### OSCAR-overexpression in joint tissue induces cartilage degeneration

Since we found recently that OA cartilage overexpresses OSCAR[14,15], we asked whether overexpressing OSCAR in naive joints would induce OA. Since Ad-OSCAR infection of chondrocytes in vitro strongly upregulates their OSCAR expression (Fig. 1a, b) and IA-injection of recombinant adenovirus effectively delivers genes to joint tissues[19,20], we IA-injected murine knees with Ad-OSCAR (3 weekly injections; Fig. 1c). By the end of the third week, this induced in vivo chondrocyte-overexpression of OSCAR (Fig. 1d), damaged the glycosaminoglycans in the articular cartilage, and induced synovitis. Notably, there was no obvious cartilage loss or osteophyte development, and the thickening of the subchondral bone plate (SBP) that is suggestive of sclerosis was not observed (Fig. 1e, f). However, as will be detailed later, when mice were treated for 8 weeks with IA Ad-OSCAR injections, cartilage destruction, osteophyte formation, SBP thickening, and synovitis were observed (Fig. 1h, i). Nonetheless, it should be noted that the Ad-OSCAR-induced model involves OA-like disease rather than classical OA.

Conversely, knocking down OSCAR in mice that were developing DMM-induced OA significantly ameliorated OA[14]: joint histology at 9 weeks of mice that underwent weekly IA-injections with small-hairpin OSCAR-expressing adenovirus (Ad-shOSCAR) starting 1 week after DMM surgery revealed marked improvements in their severe cartilage destruction (as indicated by Osteoarthritis Research Society International [OARSI] grading), osteophyte maturity, SBP thickness, and synovitis. These effects depended on Ad-shOSCAR multiplicity-of-infection (MOI) (Supplementary Fig. 1). Thus, OSCAR is necessary and sufficient for the development of OA/OA-like disease.

### Screening and verification of OSCAR-antagonists

The extracellular OSCAR domain binds to collagen-II via the GPOG-PAGFO consensus sequence[10]. Moreover, this peptide (denoted as COL^pep) binds both to recombinant OSCAR protein and cell-surface OSCAR on chondrocytes[14]. It should be noted at this point that while normal chondrocytes express OSCAR-mRNA/protein at negligible levels, we found that this expression is upregulated in vitro by IL-1β[14], which stimulates chondrocytes to produce cartilage-destroying enzymes such as MMPs and aggrecanase[6,21]. Moreover, culture of normal chondrocytes on COL^pep-coated plates also increases their native OSCAR transcription, and IL-1β augments it further (Supplementary Fig. 2a). While unstimulated Ad-OSCAR-infected chondrocytes express ~300-fold more OSCAR than IL-1β+COL^pep-stimulated uninfected chondrocytes (Supplementary Fig. 2b), IL-1β and/or COL^pep also augment it (Supplementary Fig. 2c). These OSCAR-transcription responses reflect downstream signaling induced by COL^pep-bound OSCAR that triggers new OSCAR expression[15]. These responses served as readouts of the binding of chondrocyte-surface OSCAR to COL^pep in our study.

Thus, COL^pep was used in our ELISA-based screening system to find small molecules that structurally inhibit the collagen-II binding of OSCAR. The system uses purified hOSCAR-Fc, a fusion protein of human OSCAR and human-IgG Fc, which efficiently binds to COL^pep (ref. 14). The screening involved immobilizing COL^pep, adding hOSCAR-Fc with small molecules, and detecting disruption of hOSCAR-Fc binding to COL^pep (Supplementary Fig. 2d). The 3287 screened small compounds were structurally diverse and originated from a natural-compound library and an FDA-approved drug library (Supplementary Fig. 2e). Of the 165 primary target compounds that emerged, 124 compounds were excluded because their role in chondrocytes or OA has already been studied (Supplementary Fig. 2f; Supplementary Data 1). Of the 41 remaining candidates, three underwent further study because they competed with COL^pep most strongly (Supplementary Fig. 2g). They were epigallocatechin-3-gallate (EGCG), morin hydrate, and 5-ASA (Supplementary Fig. 2h). Dose-ELISAs confirmed that these molecules suppressed hOSCAR-Fc binding to COL^pep (Supplementary Fig. 3a) and 5-ASA had the lowest IC10, IC50 and IC90 value (Supplementary Fig. 3b).

### 5-ASA competes with collagen-II for binding to OSCAR

Next, we asked whether the three candidate molecules could compete with COL^pep for binding to chondrocyte-expressed OSCAR. We thus cultured chondrocytes on COL^pep-coated plates, infected them with Ad-OSCAR and treated them with increasing candidate molecule concentrations. OSCAR-qRT-PCR showed that only 5-ASA downregulates COL^pep-induced OSCAR transcription in chondrocytes (Supplementary Fig. 3c). Since 10 μM 5-ASA had a particularly marked effect, this concentration was used further in vitro experiments.

5-ASA-mediated downregulation of OSCAR expression was also observed when OSCAR expression on normal human and murine chondrocytes was upregulated by IL-1β+COL^pep treatment: 5-ASA significantly antagonized OSCAR-protein expression in this setting (Supplementary Fig. 3d). Similarly, 5-ASA antagonized the high OSCAR-protein and mRNA expression in IL-1β+COL^pep-treated Ad-OSCAR-infected chondrocytes (Supplementary Fig. 3e, f). It should be noted here that since our screening ELISAs showed that 5-ASA binds directly to OSCAR, 5-ASA probably modulates OSCAR expression in chondrocytes by displacing COL^pep from cell-surface OSCAR, thus eliminating the downstream signaling induced by COL^pep-bound OSCAR that stimulates de novo OSCAR expression.

These results show that 5-ASA powerfully competes with OSCAR-collagen binding. Induced-fit docking (IFD) studies showed that Tyr166, Tyr200, and Ser211 in OSCAR were important for its binding to

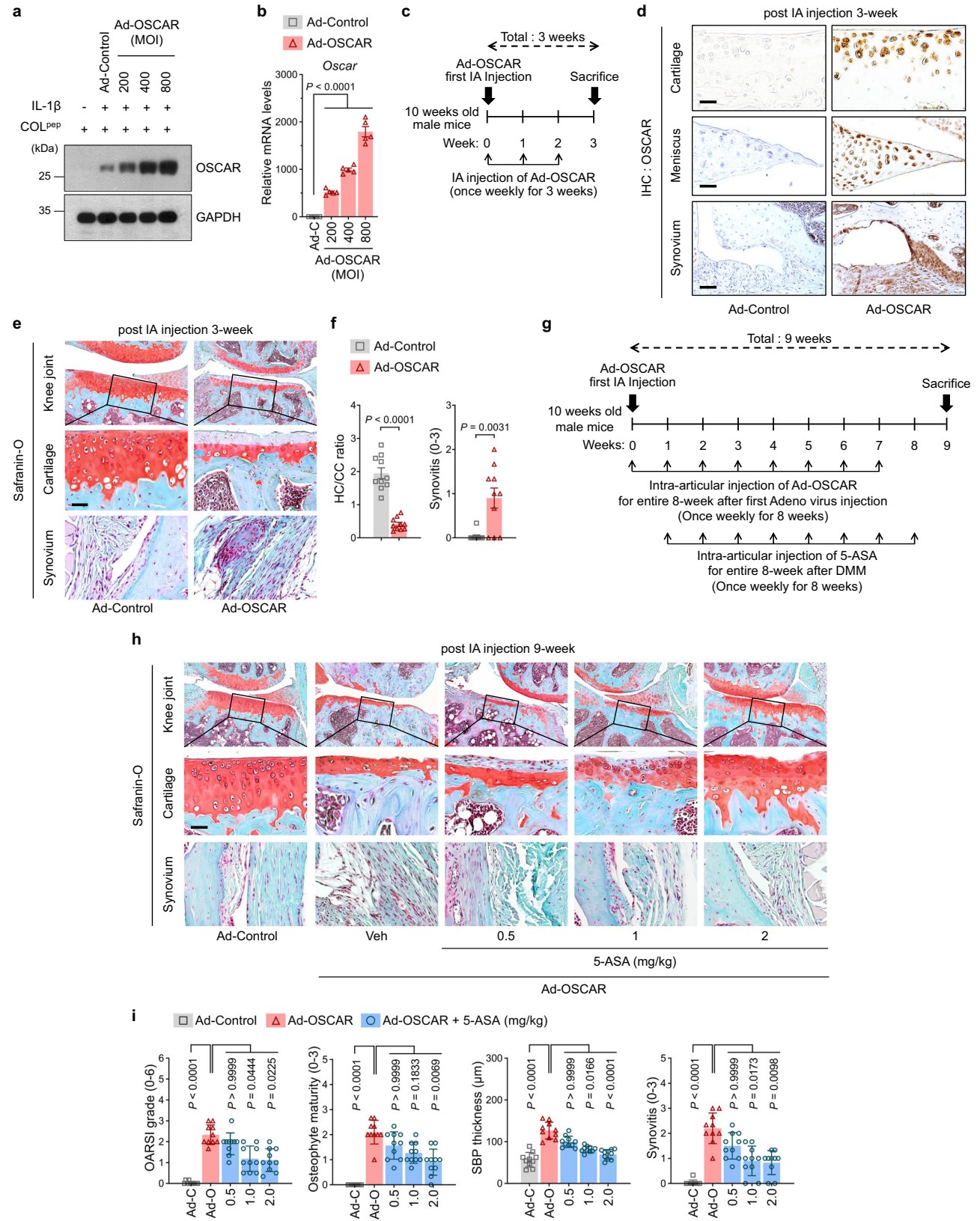

epigallocatechin-3-gallate, morin hydrate, and 5-ASA (Supplementary Fig. 3g). More specifically, the IFD analysis showed that hOSCAR and 5-ASA likely bound stably because (i) the 5-amino and carboxylate groups of 5-ASA form hydrogen bonds with Ser211 and Tyr166 in hOSCAR-Fc, respectively; and (ii) the aromatic ring of 5-ASA interacts with Tyr200 through π-π stacking. This was confirmed by mutating each residue in 5-ASA to alanine and conducting the COL^pep/hOSCAR-Fc ELISA: each substitution abolished 5-ASA binding to OSCAR (Supplementary Fig. 3h). Thus, 5-ASA is an antagonist that binds at Y166, Y200, and S211 of OSCAR and inhibits its binding to collagen-II.

**Fig. 1 | 5-ASA attenuates OA caused by OSCAR overexpression in mouse articular cartilage. a, b** Cultured primary chondrocytes from C57BL/6 J mice were infected with Ad-Control (Ad-C; 800 MOI) or the indicated MOI of Ad-OSCAR (n = 5). **a** Western blotting with anti-OSCAR antibody. **b** qRT-PCR analysis of OSCAR mRNA expression. **c–e** C57BL/6 J mice underwent IA injections once per week for 3 weeks with Ad-Control or Ad-OSCAR (both 800 MOI) and were sacrificed a week later (n = 10 mice per each group) (**c**). Cartilage sections from their knee joints were subjected to immunohistochemistry for OSCAR (**d**) and Safranin-O/hematoxylin staining (**e**). Representative images are shown. The latter slides were used to score the following OA variables (**f**): The ratio of hyaline cartilage (HC) to calcified cartilage (CC) and synovitis (n = 10 mice per group). Scale bar, 25 μm. **g–i** The knee joints of C57BL/6 J mice were IA-injected with Ad-Control or Ad-OSCAR weekly for 8 weeks and, starting 1 week after the first adenovirus injection, 0.5, 1, or 2 mg/kg

5-ASA or vehicle (Veh; PBS) was IA-injected weekly for 8 weeks starting 1 week after the first adenovirus injection. The last 5-ASA/vehicle injections occurred 1 week after the last adenovirus injection and 1 week before the sacrifice at week 9 (n = 10 mice per group) (**g**). Safranin-O/hematoxylin staining of cartilage sections (**h**) was then conducted. Representative images are shown. The slides were used to score the following OA variables (**i**): articular cartilage destruction (as indicated by the OARSI grade), osteophyte maturity, subchondral bone plate thickness (SBP Th.), and synovitis. The OARSI grade, synovitis, and osteophyte maturity data are shown as means ± 95% confidence intervals (CI). Differences between groups were determined with the Kruskal–Wallis test followed by the Mann–Whitney U test. The HC/CC ratio and SBP thickness data were shown as means ± s.e.m. Differences between groups were determined with a two-tailed t-test. Exact P values can be found in the accompanying Source Data. Scale bars, 25 μm.

## 5-ASA treatment attenuates OSCAR-related cartilage destruction in mice

Since (i) cartilage expresses more OSCAR in OA[14,15], (ii) collagen-II is the activating ligand of OSCAR[10], and (iii) 5-ASA antagonizes OSCAR binding to collagen-II, we asked whether IA-injection of 5-ASA could attenuate the OA-like cartilage destruction induced by 8 IA Ad-OSCAR injections. Indeed, when the mice were co-injected weekly with 0.5, 1, or 2 mg/kg 5-ASA starting 1 week after the first Ad-OSCAR injection and sacrificed 1 week after the last 5-ASA injection (Fig. 1g), we found that 5-ASA reduced articular cartilage erosion (i.e., OARSI scores) as well as decreasing osteophyte development, SBP thickening, and synovitis (Fig. 1h, i). Moreover, as we will detail later, we found that the cartilage destruction induced by DMM surgery, which correlates with cartilage expression of OSCAR (see Fig. 7), was greatly ameliorated by IA injections of 5-ASA (see Figs. 6 and 7). Thus, 5-ASA may antagonize the ability of OSCAR to induce cartilage destruction.

It should be noted that mice treated weekly with IA 5-ASA injections for 3 or 8 weeks were viable, normally sized, had normal life-spans, and lacked gross morphological or histological abnormalities.

## 5-ASA powerfully targets PPARγ and the eicosanoid pathway

Supplementary Fig. 3 and Fig. 1 together suggest that by binding to OSCAR, 5-ASA alters OSCAR signaling in chondrocytes, which both reduces chondrocyte OSCAR expression and associates with cartilage protection. To identify the 5-ASA targets in this signaling pathway, we conducted RNA-sequencing analysis on Ad-OSCAR-infected chondrocytes cultured on COL^pep with or without 5-ASA.

Compared to Ad-Control-infected chondrocytes, OSCAR overexpression resulted in 1515 DEGs, of which 682 were upregulated and 833 were downregulated. By contrast, compared to Ad-Control-infected chondrocytes, 5-ASA-treated OSCAR-overexpressing chondrocytes had 456 DEGs, of which 184 were upregulated and 272 were downregulated. Importantly, 5-ASA downregulated 82 of the 682 DEGs (12%) that were upregulated by OSCAR overexpression and upregulated 118 of the 833 DEGs (14%) that were downregulated by OSCAR overexpression. These DEGs were termed Flip-DEGs because 5-ASA reversed (flipped) the effect of OSCAR overexpression (Supplementary Data 2). Since Ad-OSCAR overexpression in murine knees induced cartilage destruction and OA-like disease and 5-ASA repressed that, we, therefore, designated the DEGs that were upregulated by OSCAR overexpression but then downregulated by 5-ASA as Flip^catabolic-DEGs because these genes could potentially promote cartilage destruction/OA: indeed, these DEGs included MMPs and ADAMTSs. Similarly, the DEGs that were downregulated by OSCAR overexpression but then upregulated by 5-ASA were designated Flip^anabolic-DEGs because these genes could potentially inhibit cartilage destruction/OA: indeed, these DEGs included collagen-II, ACAN, and the cartilage-anabolism marker SOX9 (Fig. 2a, b; Supplementary Data 2).

Moreover, pathway analysis showed that known cartilage catabolism and anabolism pathways in OSCAR-overexpressing chondrocytes were altered by 5-ASA treatment. Specifically, the Flip^catabolic

pathways included the inflammation and eicosanoid-related pathways (Fig. 2c). In particular, the Flip^catabolic genes in the eicosanoid-related pathway included prostaglandin-endoperoxide synthase 2 (Ptgs2, which encodes cyclooxygenase [COX]−2), arachidonate 5-lipoxygenase (Alox5, which encodes LOX-5), and their downstream genes (e.g., Ptges, Alox5ap, Ltc4s, Ltb4r1, and Ltb4r2) (Supplementary Fig. 4a). Moreover, the Flip^anabolic pathways included those that relate to collagen, NMDA-receptor and gap-junction trafficking, cholesterol-metabolism, and folate/amino-acid metabolism (Fig. 2d). Several of the latter are known to relate to cartilage-regeneration processes, including collagen production[22] and glycine/serine metabolism, which is required for the biosynthesis of ECM collagen and glycoprotein[23]. These transcriptomic changes, together with the in vitro chondrocyte, induced docking, and Ad-OSCAR-injection data, support the notion that 5-ASA may exert chondroprotective effects by altering the COL^pep-stimulated signaling of OSCAR on chondrocytes.

To assess the changes further, we identified the 14 transcription factors (TFs) whose expression was most strongly altered by 5-ASA treatment (Fig. 2e) and then conducted network analysis (Fig. 2f). This suggested that PPARγ-encoded PPARG may play an important role in the effect of 5-ASA on chondrocytes. It was downregulated by OSCAR overexpression but this was flipped by 5-ASA. This is consistent with studies showing that 5-ASA upregulates PPARγ in epithelial cells[24]. Other important TFs may be EP300, which is a co-activator of PPARγ[25]; SOX9, which is a cartilage-anabolism marker in OA[26]; SREBF1, which mediates cholesterol metabolism[27], which is one of the 5-ASA-regulated pathways (Fig. 2); ATF4, which promotes SREBF1 by inhibiting its degradation[28] and may upregulate collagen synthesis[29] and amino-acid metabolism[30]; and RXRA, which heterodimerizes with PPARγ and may thereby regulate lipid/cholesterol metabolism[31].

To further determine which FlipDEGs and FlipTFs could be particularly important 5-ASA targets in OA, we asked whether the 5-ASA-altered DEGs demonstrated similar expression patterns in response to other known drugs/compounds. For this, we conducted in silico analysis with Connectivity Map (CMap), a large dataset comprising the transcriptomes of >20 K drugs/compounds that have been used extensively for drug repurposing and mode-of-action analyses[32]. This analysis indicated how closely the 5-ASA-induced Flip-DEG profile resembled the DEG profiles induced by each of the >20 K drugs/compounds. We then took the top 10% of the most similar CMap profiles and listed the most enriched targets (see Methods for details). This revealed eight putative 5-ASA targets. Three have already been noted in the analyses above, namely, PTGS2/COX-2, RXRA, and PPARG. The other five predicted targets were ERBB2, TBXAS1, NR1I2, PPARD, and RARG (Fig. 2g): these were also in the Flip^catabolic and Flip^anabolic gene sets (Supplementary Data 2), and two (PPARD and RARG) were in the 14 TFs that we found were most strongly altered by 5-ASA treatment (Fig. 2e). Moreover, Zhu et al.[33] showed that an RXR agonist can suppress OA and that this relies on activation of PPARγ. In addition, like PTGS2/COX-2, TBXAS1 is also a key enzyme in the eicosanoid pathway (Supplementary Fig. 4a, b).

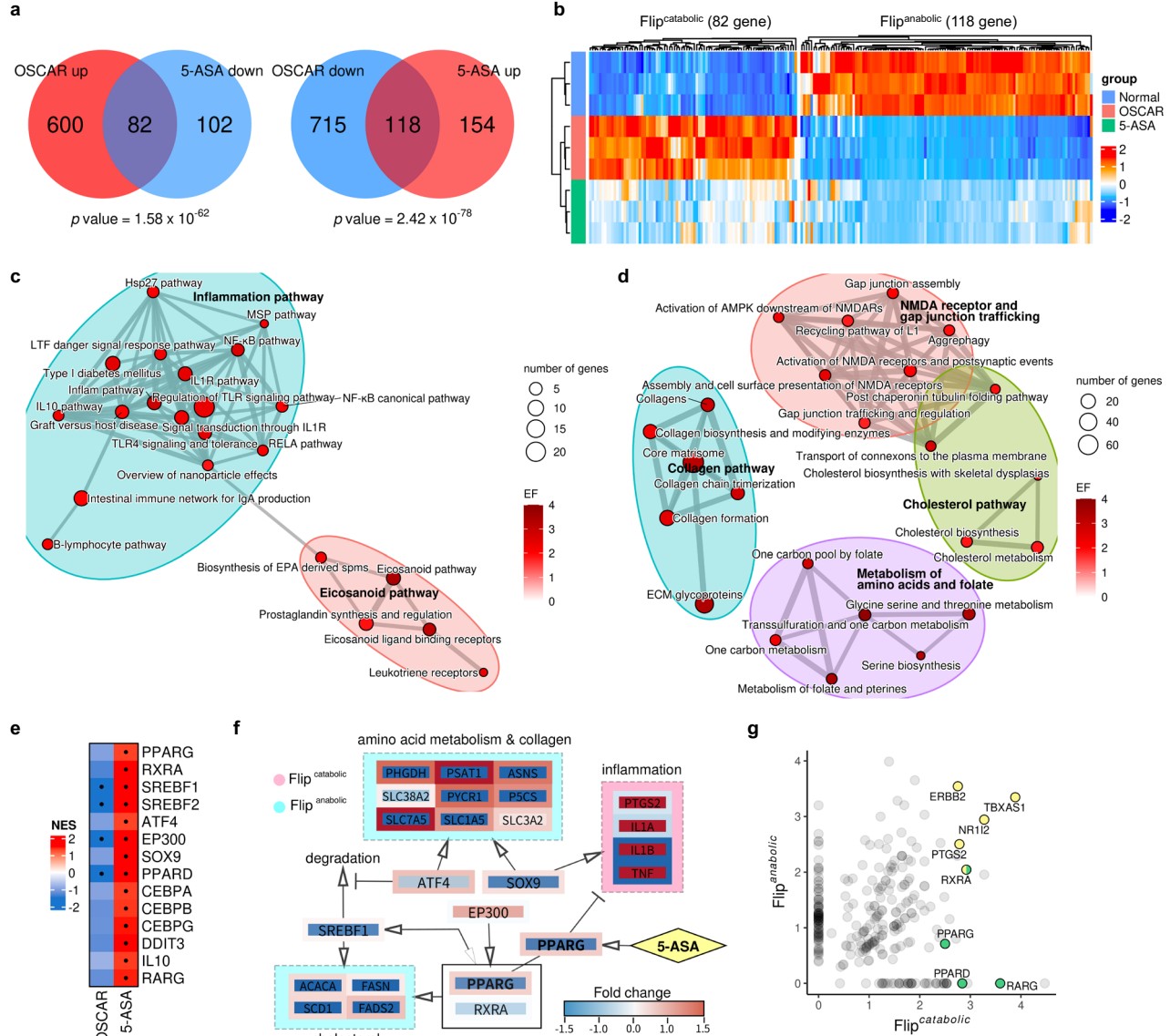

**Fig. 2 | 5-ASA ameliorates OA by regulating the transcriptomic changes induced by OSCAR. a** Identification of DEGs in Ad-OSCAR-infected chondrocytes relative to Ad-Control-infected chondrocytes and the effect of 5-ASA treatment on these DEGs. Chondrocytes infected with Ad-OSCAR were treated with or without 10 μM 5-ASA. Chondrocytes infected with Ad-Control were not treated with 5-ASA. The total RNAs were subjected to RNA-seq analysis. The 82 genes that were upregulated by OSCAR overexpression but then downregulated by 5-ASA are denoted as Flip[Up-Down] DEGs. The 118 genes that were downregulated by OSCAR overexpression but then upregulated by 5-ASA were designated Flip[Down-Up] DEGs. **b** Heatmap of the Flip[Up-Down] and the Flip[Down-Up] genes. **c**, **d** Network of the pathways that are inversely regulated ('Flip') by OSCAR and 5-ASA. The pathways are connected by significantly overlapping genes. Flip[catabolic] pathways are shown (**c**) and Flip[anabolic] pathways (**d**). Pathways are grouped with K-means clustering.

**e** Identification of the TF genes that operate downstream of the 'Flip' pathways (FlipTFs) in OSCAR-overexpressed and 5-ASA-treated chondrocytes. The normalized enrichment score (NES) for these TFs is shown. **f** Inferred regulatory network between the FlipTFs and the Flip pathway genes. Genes are colored according to fold change of OSCAR (center) and 5-ASA (border). The networks were visualized by Cytoscape v.3.8 software. **g** Identification of the 5-ASA-associated FlipDEGs and FlipTFs that are also strongly regulated by other FDA-approved drugs. The CMap database was searched for drugs that strongly altered the Flip[catabolic] and Flip[anabolic] DEGs in the same direction as 5-ASA. The Flip[Up-Down] and Flip[Down-Up] genes/TFs were then graphed on the x- and y-axis, respectively. The FlipDEGs (yellow) and FlipTFs listed in **e** (green) that are significantly enriched drug targets are highlighted. A hypergeometric test was used to conduct an enrichment analysis, assessing the extent of overlap between the two groups across the common gene space (**a**).

## PPARγ signaling in chondrocytes may mediate the cartilage-protective effects of 5-ASA

Of the DEGs/TFs that we identified, *Ptgs2*/COX-2, *Alox5*/LOX-5, and *Pparg*/PPARγ were most strongly regulated by OSCAR overexpression in chondrocytes, and these effects were vigorously reversed by 5-ASA. Thus, by downregulating PPARγ, OSCAR may upregulate COX-2 and LOX-5. These then promote inflammation because they are key arachidonic-acid metabolizers and convert it into inflammatory eicosanoids, namely, the prostaglandins (PGs), thromboxanes, leukotrienes (LTs), and lipoxins[34,35]. COX exists as COX-1 and COX-2

isoforms. The latter produces PGE2, which drives smooth muscle contraction, pain, fever, and vasodilation. Notably, specific COX-2 inhibitors have been used to treat OA and rheumatoid arthritis (RA) with a low risk of adverse gastrointestinal effects[36]. Of the six LOX genes, LOX-5 produces LT[37], which activates leukocytes. Notably, LTB4 plays an essential role in RA-associated pain and bone damage[38].

Since elucidating the molecular mechanisms by which 5-ASA protects cartilage could reveal therapeutic targets, we focused further on COX-2 and LOX-5 as either markers of the eicosanoid pathway or true targets. In either case, exploring the link between OSCAR, PPARγ,

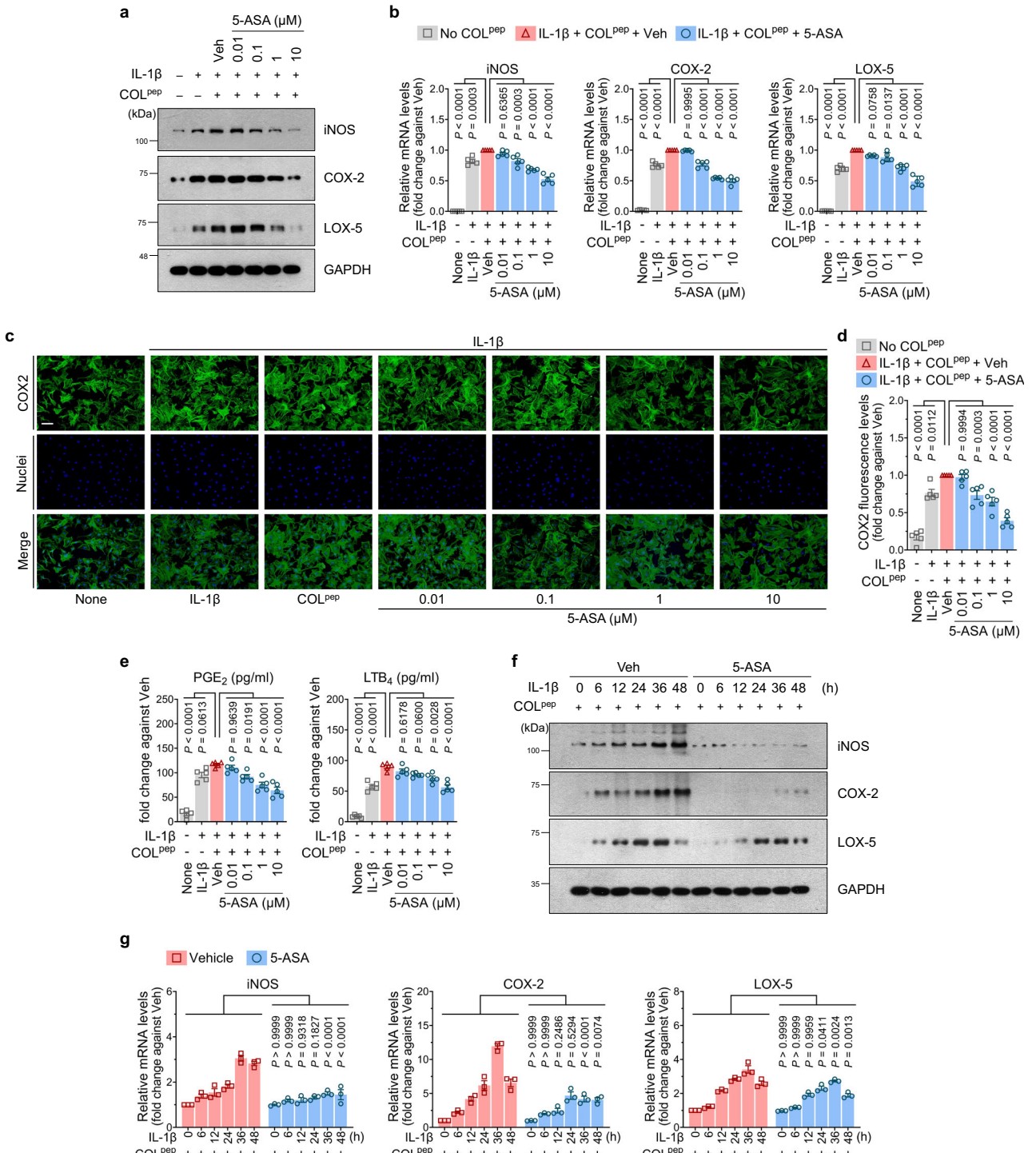

**Fig. 3 | 5-ASA inhibits the PPARγ-COX-2 pathway in primary chondrocytes.**
**a**–**e** Ad-OSCAR-infected mouse chondrocytes were cultured for 48 h on dishes that were coated with 2 μg/ml COL^pep or the GPP10 control peptide together with 10 ng/ml IL-1β and either vehicle or the indicated concentration of 5-ASA (n = 5 independent primary chondrocyte cultures). **a** Western blotting with anti-iNOS, -COX-2, and -LOX-5 antibodies. **b** qRT-PCR analysis of iNOS, COX-2, and LOX-5 mRNA. **c, d** Immunofluorescence staining of COX-2. **c** COX-2 was stained green (upper boxes), while the nuclei were blue due to DAPI staining (middle boxes). The merged images are shown in the lower boxes. (Scale bar, 50 μm). **d** Quantification

of the fluorescence intensity in each group. **e** ELISA analysis of PGE2 and LTB4 concentrations released into the supernatant. **f, g** Primary mouse chondrocytes were cultured with 10 ng/ml IL-1β and COL^pep and iNOS, COX-2, and LOX-5 immunoblotting (**f**) and qRT-PCR (**g**) were performed (n = 5 independent primary chondrocyte cultures). The data are shown as means ± s.e.m. P-values were determined by one-way ANOVA followed by Tukey's multiple comparisons test (**b, d, e**) or two-way ANOVA followed by Sidak's post hoc test (**g**). Source data are provided as Source Data files.

COX-2, and LOX-5 could help illuminate the key role of PPARγ in OA pathogenesis. Thus, we asked whether OSCAR binding to collagen-II affected the eicosanoid pathway in chondrocytes. Indeed, IL-1β +COL[pep]-treated Ad-OSCAR-infected chondrocytes demonstrated *Ptgs2*/COX-2 and LOX-5 protein/mRNA upregulation. Inducible nitric oxide synthase (iNOS), which produces the key inflammatory mediator nitric oxide[39], was also upregulated. Significantly, 5-ASA treatment inhibited all these effects (Fig. 3a, b). Immunofluorescence staining confirmed that IL-1β+COL[pep] upregulated *Ptgs2*/COX-2, and 5-ASA suppressed this effect (Fig. 3c, d). As expected, these changes also affected chondrocyte PGE2 and LTB4 expression: IL-1β+COL[pep] elevated the secretion of both mediators into the supernatant, and 5-ASA suppressed this (Fig. 3e). We then confirmed that 5-ASA down-regulated *Ptgs2*/COX-2 expression via OSCAR by infecting chondrocytes with Ad-Ptgs2 and then treating them with IL-1β+COL[pep] with or without 5-ASA: 5-ASA significantly reduced the overexpression of *Ptgs2*/COX-2. This was not observed when IL-1β+COL[pep] was not present, which indicates that 5-ASA suppressed *Ptgs2*/COX-2 expression via OSCAR (Supplementary Fig. 5a, b). The gain-of-function experiments conducted with COX2 via Ad-Ptgs2 revealed that the over-activation of the eicosanoid pathway results in an opposing effect when compared to the impact of 5-ASA. The overexpression of Ptgs2 exhibited a contrasting effect to that of 5-ASA concerning the expression of Flip[catabolic] DEGs, including well-known markers of carti-lage catabolism, such as *Mmp3*, *Mmp9*, *Mmp13*, and *Adamts5* mRNA, in chondrocytes (Supplementary Fig. 5c). The possibility that 5-ASA blocked COL[pep]-induced OSCAR-mediated PGE2 production via *Ptgs2*/COX-2 was confirmed by infecting chondrocytes with Ad-shPtgs2, which silences *Ptgs2*/COX-2: when these cells were stimulated with COL[pep], they were unable to produce PGE2, as expected, and 5-ASA treatment had no effect on this (Supplementary Fig. 5d, e). IL-1β +COL[pep] treatment of primary (uninfected) chondrocytes also elevated COX-2, LOX-5, and iNOS protein/transcripts, and these effects were reversed by 5-ASA (Fig. 3f, g).

Since PPARγ can inhibit COX-2 expression[40] and *Ptgs2*/COX-2 expression was strongly suppressed by 5-ASA, we speculated that COL[pep]-stimulated OSCAR promotes *Ptgs2*/COX-2 by downregulating PPARγ and that 5-ASA relieves this suppressive effect by competing with COL[pep]. To test this, IL-1β+COL[pep]-treated primary chondrocytes were incubated with either 5-ASA or the PPARγ-agonist rosiglitazone[41], and PPARγ expression was measured. As expected, IL-1β+COL[pep] downregulate PPARγ expression. However, 5-ASA greatly increased it, in fact, even better than rosiglitazone (Fig. 4a). Thus, 5-ASA binding to OSCAR strongly induced PPARγ. Next, we induced chondrocytes to overexpress PPARγ with Ad-Pparg (Fig. 4b), cultured them with IL-1β +COL[pep] to upregulate OSCAR expression, treated them with 5-ASA, and measured *Ptgs2*/COX-2 expression. While IL-1β+COL[pep] increased *Ptgs2*/COX-2 expression as expected, overexpressing PPARγ halved that effect, and this was further augmented when 5-ASA was also present (Fig. 4c). Moreover, overexpressing *Pparg* had the same effect as 5-ASA in terms of the Flip[anabolic] DEGs (and known markers of carti-lage anabolism[7,8]) *Rxra*, *Col2a1*, *Acan*, and *Sox9* mRNA expression by the IL-1β+COL[pep]-treated chondrocytes: both treatments increased these mRNA levels (Supplementary Fig. 5f). These results together suggest that (i) when OSCAR expression in chondrocytes is increased by collagen binding, PPARγ is downregulated, which upregulates *Ptgs2*/COX-2 expression and thereby elevates PGE2 secretion and inflammation; and (ii) 5-ASA treatment reverses this pro-inflammatory effect on the PPARγ-COX-2-PGE2 axis.

### 5-ASA stimulates MSC chondrogenesis
The key role of PPARγ in the mode-of-action of 5-ASA in chondrocytes suggested another possibility by which 5-ASA improves OA, namely, it may promote cartilage regeneration. This notion reflects the fact that (i) MSCs from OA patients differentiate into chondrocytes in vitro less

well than normal-donor MSCs[42], and (ii) PPAR-γ promotes human-MSC chondrogenesis in vitro[43]. To test whether 5-ASA promotes chon-drogenesis by blocking OSCAR activity, we first asked whether OSCAR could impair MSC chondrogenesis. Thus, the damaged/undamaged articular cartilage areas of ten OA patients who underwent total knee-replacement surgery (Supplementary Data 3) were identified with Alcian-Blue staining (which recognizes cartilage glycosaminoglycans). The damaged/undamaged areas were then subjected to OSCAR and PPARγ immunohistochemistry. Significantly, OSCAR protein was very low in the undamaged cartilage but highly upregulated in the damaged cartilage, whereas PPARγ protein demonstrated the opposite pattern (Fig. 4d). Thus, high OSCAR and low PPARγ expression could poten-tially be associated with poor cartilage regeneration and MSC activity.

Next, we asked whether 5-ASA treatment promoted murine bone marrow-derived MSC differentiation into glycosaminoglycan-producing chondrocytes. We noted that after 21 days of culture in a chondrogenic medium containing COL[pep], the differentiating chon-drocytes started to express OSCAR. This was suppressed when 5-ASA was also present (Fig. 4e). Notably, 5-ASA also increased chondrogen-esis in a dose-dependent manner, as shown by greater Alcian-Blue staining (Fig. 4f). Moreover, this not only associated with decreased *Oscar* transcription, it also associated with increased expression of *Pparg*, the Flip[anabolic] DEG *Rxra*, and the three markers cartilage-anabolism genes collagen-II/*Col2a1*, *Acan*, and *Sox9*[26] (Fig. 4g). The expression of the latter indicates that chondrocytes emerging in the presence of 5-ASA are capable of cartilage anabolism. Interestingly, if the bone marrow-derived MSCs were infected with Ad-shPparg before culture in chondrogenic medium containing COL[pep], 5-ASA could no longer increase cell expression of *Col2a1*, *Acan*, and *Sox9* (Supplemen-tary Fig. 5g). This suggests that (i) PPARγ can induce beneficial cartilage anabolism (as well as antagonize harmful eicosanoid pathway-mediated joint inflammation), and (ii) 5-ASA can promote the production of cartilage-anabolic chondrocytes in a PPARγ-dependent manner.

The pro-chondrogenic effect of 5-ASA was also observed when human chondrocytes underwent pellet or 3-dimensional culture with soluble COL[pep] with/without 5-ASA: 5-ASA clearly associated with greater pellet diameters (Fig. 4h), chondrocyte cohesion, and healthy growth (Fig. 4i). Thus, OSCAR expression associated with cartilage damage and suppressed PPARγ expression, while 5-ASA promoted MSC chondrogenesis in a manner that relied on increased PPARγ expression. This suggests that 5-ASA could potentially promote carti-lage regeneration.

### 5-ASA may prevent cartilage destruction by blocking OSCAR-mediated upregulation of ECM degradation
OA chondrocytes demonstrate upregulation of the ECM-degrading enzymes MMP3, MMP9, MMP13, and ADAMTS5[4,5], which are crucial effectors of the cartilage destruction in OA[44], and downregulation of the ECM molecules collagen-II and ACAN[45]. These changes are thought to be caused by the pro-inflammatory molecule IL-1β[4,5]. Notably, our RNA-seq analysis showed that OSCAR overexpression strongly upre-gulated MMP3, MMP9, MMP13, and ADAMTS5, and downregulated collagen-II, ACAN, and the cartilage-anabolism marker SOX9, and this was powerfully reversed by 5-ASA treatment. This was confirmed by immunoblotting and RT-PCR of primary chondrocytes treated with IL-1β+COL[pep] or IL-1β alone (Supplementary Fig. 6).

Notably, when IL-1β-stimulated chondrocytes were treated with 5-ASA for up to 30 min, we observed that it had no effect on the strong but transient IL-1β-induced increase in MAPK and NF-κB signaling (Supplementary Fig. 7a, b). This suggests that 5-ASA ameliorates OA by reversing the longer-term OSCAR-associated pro-inflammatory signaling generated by IL-1β rather than by modulating other signaling pathways in OA. Thus, OSCAR may promote cartilage catabolism and inhibit cartilage anabolism, and 5-ASA may reverse this.

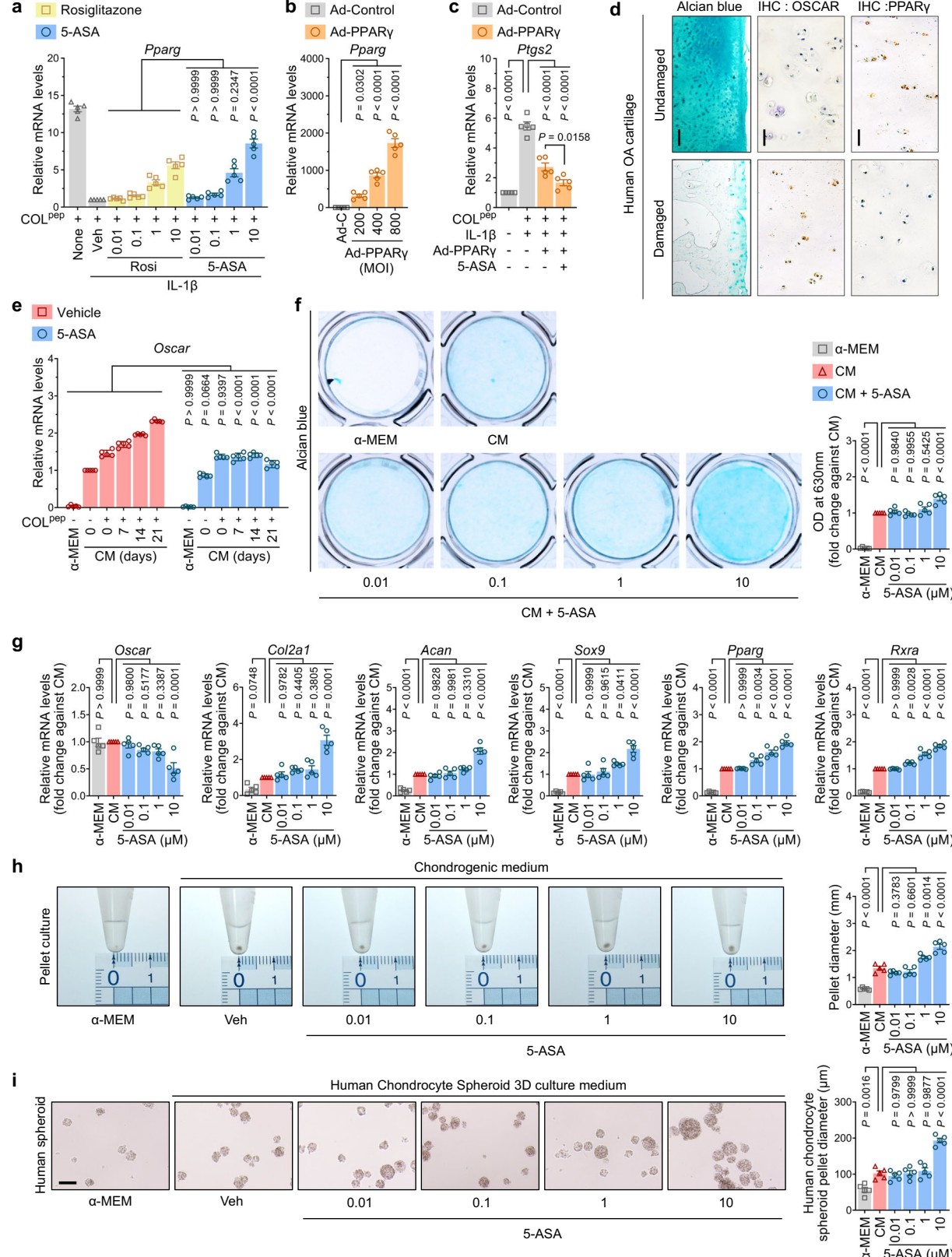

## 5-ASA treatment attenuates DMM surgery-induced OA, even when administered well after disease onset

We next asked whether IA injection of 0.5, 1, or 2 mg/kg 5-ASA attenuated DMM surgery-induced OA in mice. Thus, 5-ASA was injected for 8 weeks starting 1 week after surgery and the mice were sacrificed at 9 weeks (Fig. 5a). 5-ASA strongly inhibited articular-cartilage erosion

and other pathological signs in the affected knee in a dose-dependent fashion (Fig. 5b, c). This was confirmed by contrast agent-enhanced cartilage 3D micro-computed tomography: 5-ASA markedly inhibited SBP thickening and bone-volume increase (Fig. 5d, e). Thus, local 5-ASA administration significantly reduced trauma-induced articular-cartilage degeneration.

**Fig. 4 | 5-ASA enhances PPARγ signaling in chondrocytes and chondrogenesis.**
**a** Primary chondrocytes were treated without IL-1β or with 10 ng/ml IL-1β and 10 μM 5-ASA or 10 μM rosiglitazone (Rosi), and *Pparg* mRNA was measured with qRT-PCR. **b** Primary chondrocytes were infected with Ad-Control (Ad-C; 800 MOI) or the indicated Ad-PPARγ MOI for 2 h, and *Pparg* mRNA was measured with qRT-PCR. **c** Primary chondrocytes were infected with 800 MOI Ad-PPARγ and then treated with 10 ng/ml IL-1β with or without 5 μM 5-ASA. COX-2/*Ptgs2* mRNA was determined by qRT-PCR (*n* = 5 independent primary cultures in **a**–**c**). **d** Articular cartilage from ten patients obtained during total knee replacement surgery was subjected to Alcian-Blue staining (left) and IHC for OSCAR and PPARγ expression (middle and right, respectively). The damaged and undamaged sections were assessed separately. Representative Alcian blue-stained (scale bar, 100 μm) and IHC (scale bar, 50 μm) images are shown. **e** Murine bone marrow-derived MSCs were cultured in a chondrogenic medium (CM) with or without 5-ASA for 7, 14, and 21 days, and qRT-

PCR was conducted to determine *Oscar*. **f**–**g** Bone marrow-derived MSCs were cultured in a chondrogenic medium with or without 5-ASA for 3 weeks, stained with Alcian Blue, and photographed (left, **f**). Alcian-Blue activity was quantified by measuring the absorbance at 630 nm (right, **g**). qRT-PCR was conducted to determine *Oscar*, *Col2a1*, *Acan*, *Pparg*, *Sox9*, and *Rxra* mRNA expression (**g**). **h** Bone marrow-derived MSCs were subjected to pellet culture in e-tubes for 3 weeks with or without 0.01, 0.1, 1, and 10 μM 5-ASA, after which the pellet diameter was measured with a micro ruler (*n* = 5 independent primary cultures in **e**–**h**). **i** Human chondrocytes were subjected to spheroid culture in a 3D microenvironment with or without 0.01, 0.1, 1, and 10 μM 5-ASA for 1 week, and the diameter of the spheroids was measured (*n* = 5 independent cultures in **i**). The data are shown as means ± s.e.m. *P*-values were obtained by one-way ANOVA followed by Tukey's multiple comparisons test (**b**, **c**, **f**–**i**) or two-way ANOVA followed by Sidak's post hoc test (**a**, **e**). Exact *P*-values were provided in the Source Data file.

Next, to determine whether 5-ASA can reduce OA when given well after OA is evident, 5-ASA treatment was only started 5 weeks after DMM surgery (Fig. 5f). In the untreated group, severe cartilage erosion, accompanied by the development of osteophytes and thickening of the subchondral bone plate, along with notable synovitis, were observed. Conspicuously, the administration of 5-ASA during weeks 5–8 demonstrated a significant amelioration of these OA parameters (Fig. 5g, h). Notably, the 5-ASA-induced reduction in disease was as large as when 5-ASA was administered for the whole 8 weeks (Fig. 5b, c). The patterns were also observed for the other markers of OA (Fig. 5).

We then showed that 5-ASA could even induce DMM surgery-damaged cartilage to recover when it was first started after the disease had progressed to severe degenerative OA, namely, 8 weeks after DMM (Fig. 6a). Hence, mice subjected to a 4-week treatment with 5-ASA from weeks 8–11 exhibited a decrease in several OA markers when contrasted with mice at the 12-week post-DMM surgery stage. In fact, the final OARSI grade of the 5-ASA-treated mice sacrificed at week 12 was much lower than that of untreated mice that were sacrificed at week 8. This was also true for the other OA indicators. Significantly, the treated mice that were killed at week 12 had thicker cartilage than the mice that were sacrificed at week 8 (Fig. 6b, c). These findings suggest that 5-ASA could not just arrest OA progression; it could reverse it. However, additional studies are needed to confirm this.

**5-ASA treatment in OA protects the cartilage from destruction and is associated with the downregulation of OSCAR and ECM-degrading enzymes**
Immunohistochemistry of the knee cartilage of mice with DMM-induced post-traumatic OA showed that the surgery upregulated OSCAR protein expression, as expected[14,15] and 5-ASA reversed this (Fig. 7a). Moreover, DMM surgery significantly upregulated MMP3, MMP13, and ADAMTS5 and downregulated COL2A1 and ACAN, and these effects were all markedly improved by 5-ASA treatment (Fig. 7). The latter findings are consistent with our RNA-seq, RT-PCR, and immunoblotting analyses.

## Discussion
The OA hallmark is the tipping of the fine balance between chondrocyte production of ECM and chondrocyte-mediated ECM catabolism: the latter predominates, which results in cartilage destruction. This reflects that enzymatic degradation, oxidative stress, and mechanical stress[46] lead to the production of ECM fragments (including COL^pep^), which are recognized as damage-associated molecular patterns (DAMPs) by receptors on chondrocytes. The DAMP signals also induce proinflammatory cytokines that promote ECM catabolism and interrupt ECM anabolism. These factors together decrease chondrocyte numbers, which reduces their ability to maintain cartilage homeostasis[47].

OSCAR may play an important role in these osteoarthrogenic mechanisms because it recognizes the triple-helix collagen peptide[10–12]. This notion was advanced by our previous study showing that OA cartilage in humans and mice expresses more OSCAR than normal cartilage[14,15]. The present study further supported this hypothesis since we found that simply overexpressing OSCAR in articular joints led to OA-like cartilage destruction and inflammation. Moreover, downregulating OSCAR expression by IA-injection of shOSCAR-expressing adenovirus significantly ameliorated DMM-induced OA (Supplementary Fig. 1). Thus, OSCAR may play a very important role in OA pathogenesis.

The putative key role of OSCAR in OA, in turn, suggests that OSCAR antagonists of OSCAR-collagen binding could suppress OA-associated articular cartilage destruction in humans. Here, we report our discovery that 5-ASA can serve as an OSCAR antagonist that effectively reduces DMM-induced OA and the cartilage destruction generated by OSCAR overexpression in the joint. It should be noted that during the writing of this paper, Li et al.[48] reported that culturing human osteochondral explant models of OA with 5-ASA downregulated cartilage degradation. While they did not link this effect to OSCAR or conduct in vivo experimentation, these findings are highly consistent with our own, which together suggest that 5-ASA may have marked chondroprotective effects.

5-ASA is an anti-inflammatory drug that is widely used to treat IBDs, especially UC[49]. Here we show that it may also be suitable as a DMOAD since it not only halted cartilage destruction in an animal model of OA, it also appeared to induce cartilage regeneration: even when 5-ASA was first administered in late-stage (8–11-week) murine OA, when there is immense cartilage damage, the OARSI grades recovered markedly and the cartilage thickness was greater than that of untreated mice that were sacrificed at week 8. Our mechanistic analyses suggest that these therapeutic properties are due to the ability of 5-ASA to bind to OSCAR, thereby triggering OSCAR-PPARγ signaling that simultaneously (i) antagonizes inflammation and (ii) promotes chondrogenesis. With regard to the former, we found that 5-ASA competed with collagen for binding to OSCAR, and its binding to OSCAR upregulated PPARγ. This downregulated the eicosanoid pathway, including COX-2 and its generation of PGE2 (Supplementary Fig. 8). The human osteochondral explant study of Li et al.[48] also observed that culture of the explants with 5-ASA downregulated COX-2. It should be noted that while it is widely believed that systemic administration of COX-2 inhibitors (i.e., common nonsteroidal anti-inflammatory drugs) does not associate with chondroprotective effects in OA, a recent systematic review suggested that some COX-2 inhibitors could act as DMOADs if they are injected IA[50]. Moreover, another recent systematic review found that even when injected systemically, some COX-2 inhibitors (e.g., celecoxib) can modify the cartilage, bone, and synovial disease in OA, and these effects are mediated by the regulation of prostaglandins and direct changes to the tissues[51]. Since our 5-ASA treatment in mice involves IA injections and is

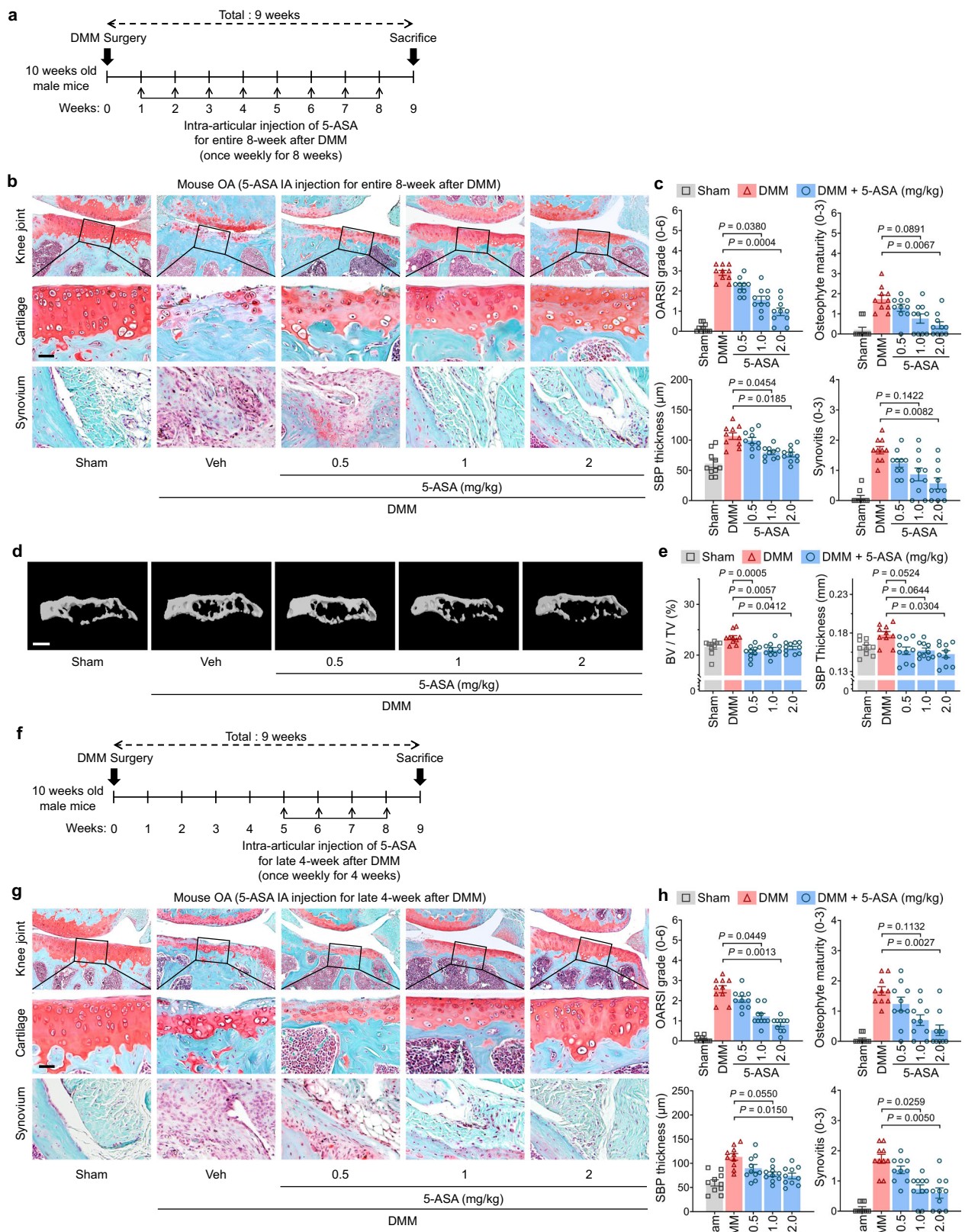

potentially a novel drug for OA, it is thus possible that COX-2 and LOX-5 could be effectors of OA. Alternatively, they are simply markers of the eicosanoid pathway. With regard to the potential role of OSCAR-PPARγ signaling in chondrogenesis, we showed that 5-ASA may also improve OA by enhancing MSC chondrogenesis, chondrocyte aggregation and growth, and cartilage repair, which is a highly desirable attribute as a

DMOAD. It is likely that PPARγ also participates in this regenerative outcome since it was upregulated by 5-ASA during chondrogenesis in vitro, and both 5-ASA treatment and overexpressing PPARg caused the developing chondrocytes to gain cartilage anabolic properties.

The importance of the OSCAR-PPARγ signaling pathway in OA is supported by the known roles of PPARγ, which is a ligand-activated TF

**Fig. 5 | 5-ASA attenuates post-traumatic OA. a–c** C57BL/6 J mice were subjected to sham operation or DMM surgery and then IA-injected once per week for the 8-week observation period with vehicle (Veh) or 5-ASA (0.5, 1, or 2 mg/kg) and sacrificed 9 weeks after the operation (*n* = 10 mice per each group) (**a**). **b** Representative Safranin-O staining and fast green counter staining images. Scale bar, 25 μm. **c** Quantitation of the following OA variables: OARSI grade, osteophyte maturity, subchondral bone plate thickness, and synovitis. **d, e** Representative reconstructed micro-CT images and quantitative analysis of mouse SBPs 10 weeks after DMM surgery compared to sham controls (**d**). Scale bars, 1000 μm.

**e** Quantitation of the SBP variables, namely, SBP thickness and bone volume fraction (BV/TV). *n* = 10 mice per each group. **f–h** C57BL/6 J mice were injected as described in (**a**), except the 5-ASA injections started at 5 weeks (**f**). Representative images are shown (**g**). OA variables were quantitated (**h**) as described in (**c**). The OARSI grade, synovitis, and osteophyte maturity data are shown as means ± 95% confidence intervals (CI). Differences between groups were determined with the Kruskal–Wallis test followed by the Mann–Whitney *U* test. Means ± s.e.m. with two-tailed *t*-test for SBP thickness. Exact *P* values can be found in the accompanying Source Data. Scale bars, 25 μm.

that plays key roles in inflammation, fibrosis, and tissue repair[52]. Moreover, PPARγ agonists protect animal models from OA, and these effects are mediated by their anti-inflammatory properties in vitro and in vivo[52]. In addition, Zhu et al.[33] showed recently that a retinoic acid metabolism blocking agent can suppress DMM-induced inflammatory signaling and OA and that this depends on the activation of PPARγ. However, the exact molecular mechanisms by which such PPARγ agonists improve OA have not been determined. Vasheghani et al.[53] also showed that mice in which PPARγ was inducibly knocked out developed very severe accelerated DMM surgery-induced OA. Thus, our study greatly advances our knowledge about the in vivo role of PPARγ in articular cartilage homeostasis, as well as confirming that upregulating this molecule may be a viable therapeutic target for OA.

Our finding that 5-ASA protected joints from OA by promoting the anti-inflammatory properties of PPARγ is consistent with UC research: multiple studies show that 5-ASA largely improves UC by increasing intestinal-wall expression of PPARγ, which downregulates local cytokine and inflammatory-mediator production[54]. The mode-of-action of 5-ASA in UC and OA exhibits similarities, which aligns with the observation that these two diseases share several pathogenic changes[55].

Thus, IA delivery of 5-ASA may have considerable therapeutic potential in OA. This also raises the possibility that systemic administration of 5-ASA that is not formulated for bowel selectivity could be useful for OA that involves multiple joints. The feasibility of 5-ASA as a DMOAD is further supported by the fact that it and its precursors (e.g., sulfasalazine, also known as azulfidine) or derivatives (e.g., 4-aminosalicylic acid, acetylsalicylic acid, methyl salicylate, and 4-aminobenzoic acid[56]) are already being used for human therapy. For example, sulfasalazine is both the first 5-ASA to be widely used for IBD and an effective disease-modifying anti-rheumatic drug for RA[57]. Moreover, oral and topical 5-ASA has undergone years of formulation development with large-scale clinical trials and is very safe[58].

In summary, our study shows that 5-ASA has chondroprotective effects that significantly ameliorate OA by modulating the OSCAR-PPARγ axis. Studies with other OA models and humans are warranted. Notably, OSCAR is a highly complex and multifactorial mediator: our RNA-seq analyses showed that OSCAR also downregulates several other signaling pathways, and 5-ASA reverses this. Moreover, PPARγ agonism can involve non-inflammatory pathways[52,59], and 5-ASA also modulates IBD by non-PPARγ pathways[60]. Thus, further research on the roles of these pathways in the protective effects of 5-ASA on articular cartilage is also needed. Research on the mechanisms by which OSCAR overexpression alone drives OA pathogenesis is also warranted.

## Methods
### Mice
Murine experiments were conducted with male 10–11-week-old C57BL/6 J (C57BL/6BomTac, DBL, South Korea) or Institute of Cancer Research (ICR) mice (IcrTac; ICR, DBL, South Korea). All mice were housed in pathogen-free barrier facilities at 5 or less per cage at 24–26 °C, humidity ranging from 30–60%, and with a 12 h light/dark cycle. The mice were randomly allocated to each experimental group. In accordance with the ethical guidelines for animal research, we

rigorously implemented animal welfare monitoring and euthanasia practices throughout the study. All animal experiments were approved by the Institutional Animal Care and Use Committees (IACUC, Protocol No: IACUC 18–109, 20–047, and 21–076) of Ewha Womans University and followed the National Research Council and ARRIVE Guidelines.

### Human samples
Cartilage tissues with OARSI grade 6 and the surrounding normal cartilage were acquired during total knee replacement surgery from ten patients with OA. The patients were 63–78 years old, and three and seven were male and female, respectively (Supplementary Data 3). To rule out the effects of other underlying diseases, we ensured that none of the patients had RA, metabolic disease, or other inflammatory diseases at the time of surgery. The institutional review board of Yonsei University (Protocol No. IRB 3-2018-0251), Gangnam Severance Hospital, South Korea, approved the use of the articular cartilage. All participants provided written informed consent authorizing the utilization of their tissue samples for research purposes and the publication of information that could potentially identify individuals.

### Recombinant adenoviruses and 5-ASA
Vector Biolabs (Malvern, PA, USA) manufactured the adenoviruses that expressed OSCAR (Ad-OSCAR; catalog No. ADV-267721), Ad-Control (1060), Ad-shControl (1122), shOSCAR (Ad-shOSCAR; shADV-267721), COX-2 (Ad-Ptgs2; ADV-281024), shCOX-2 (Ad-shPtgs2; shADV-281024), PPARγ (Ad-PPARγ; ADV-269122), or shPPARγ (Ad-shPparg; shADV-269122). 5-ASA was obtained from Sigma-Aldrich (Cat# A3537; St Louis, MO, USA) and dissolved in PBS.

### Adenovirus-induced murine OA and treatment with 5-ASA
Experimental OA was induced in C57BL/6 J WT by IA injection of Ad-OSCAR. Thus, the IA space of the right knee joint was injected along the patellar tendon with $1 \times 10^9$ plaque-forming units of Ad-OSCAR in phosphate-buffered saline (PBS; pH 7.4) (total volume, 10 μl) once a week. Mice injected with Ad-Control or the PBS vehicle served as controls. In one experiment, the weekly adenovirus injections were conducted 3 times and the mice were sacrificed a week later at week 3. In the second experiment, the adenovirus injections were conducted 8 times, and 8 weekly IA injections with 0.5, 1, or 2 mg/kg 5-ASA were also given starting 1 week after the first adenovirus injection. The mice were sacrificed at week 9.

### DMM surgery-induced murine OA and treatment with Ad-shOSCAR or 5-ASA
DMM surgery-induced OA was induced in C57BL/6 J WT male mice by surgically removing the medial meniscus ligament from the right knee joint of the hind limb[18,19,61,62]. Male mice were selected for the experiments due to a higher occurrence of the posttraumatic osteoarthritis model in males compared to females in murine studies[63]. Female hormones exhibit protective effects on cartilage, whereas male hormones exacerbate the condition[64]. Sham-operated mice served as controls. The sham operation involved conducting the same surgery on the contralateral knee but without removing the medial meniscus ligament. The mice were sacrificed 9 weeks after surgery. In one

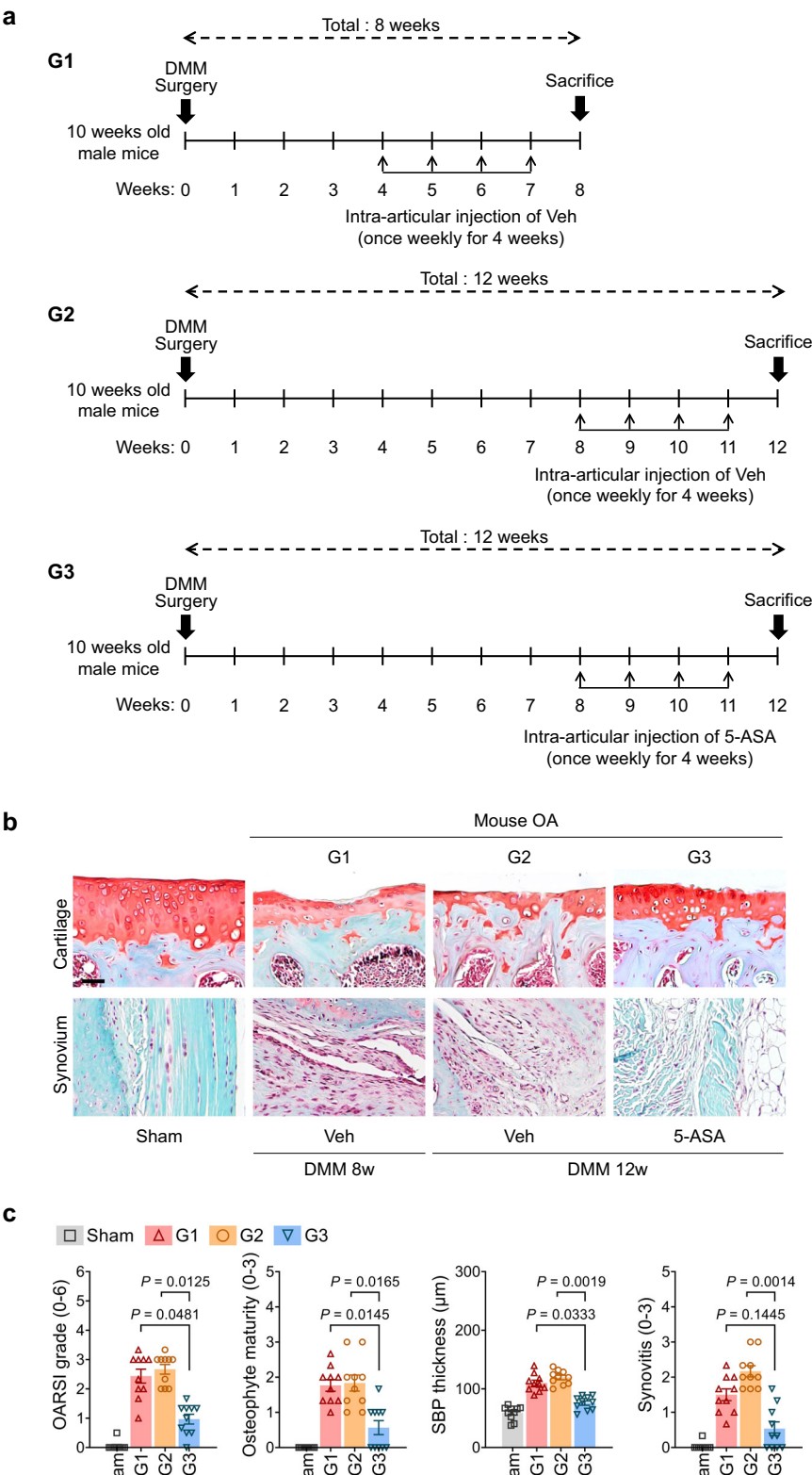

**Fig. 6 | 5-ASA induces cartilage regeneration, even when administered in late-stage OA. a** Mice were subjected to sham operation or DMM surgery and then IA-injected once per week for the indicated 4-week prior to sacrifice with vehicle (Veh; G1 and G2) or 5-ASA (2 mg/kg; G3) and sacrificed 9 (G1) or 12 weeks (G2 and G3) after the operation (*n* = 10 mice per group). **b** Representative Safranin-O staining and fast green counter staining images. Scale bar, 25 μm. **c** Quantitation of the following OA variables: OARSI grade, osteophyte maturity, subchondral bone plate thickness, and synovitis. The OARSI grade, synovitis, and osteophyte maturity data are shown as means ± 95% confidence intervals (CI). Differences between groups were determined with the Kruskal–Wallis test followed by the Mann–Whitney *U* test. Means ± s.e.m. with two-tailed *t*-test for SBP thickness. Exact *P* values can be found in the accompanying Source Data. Scale bars, 25 μm.

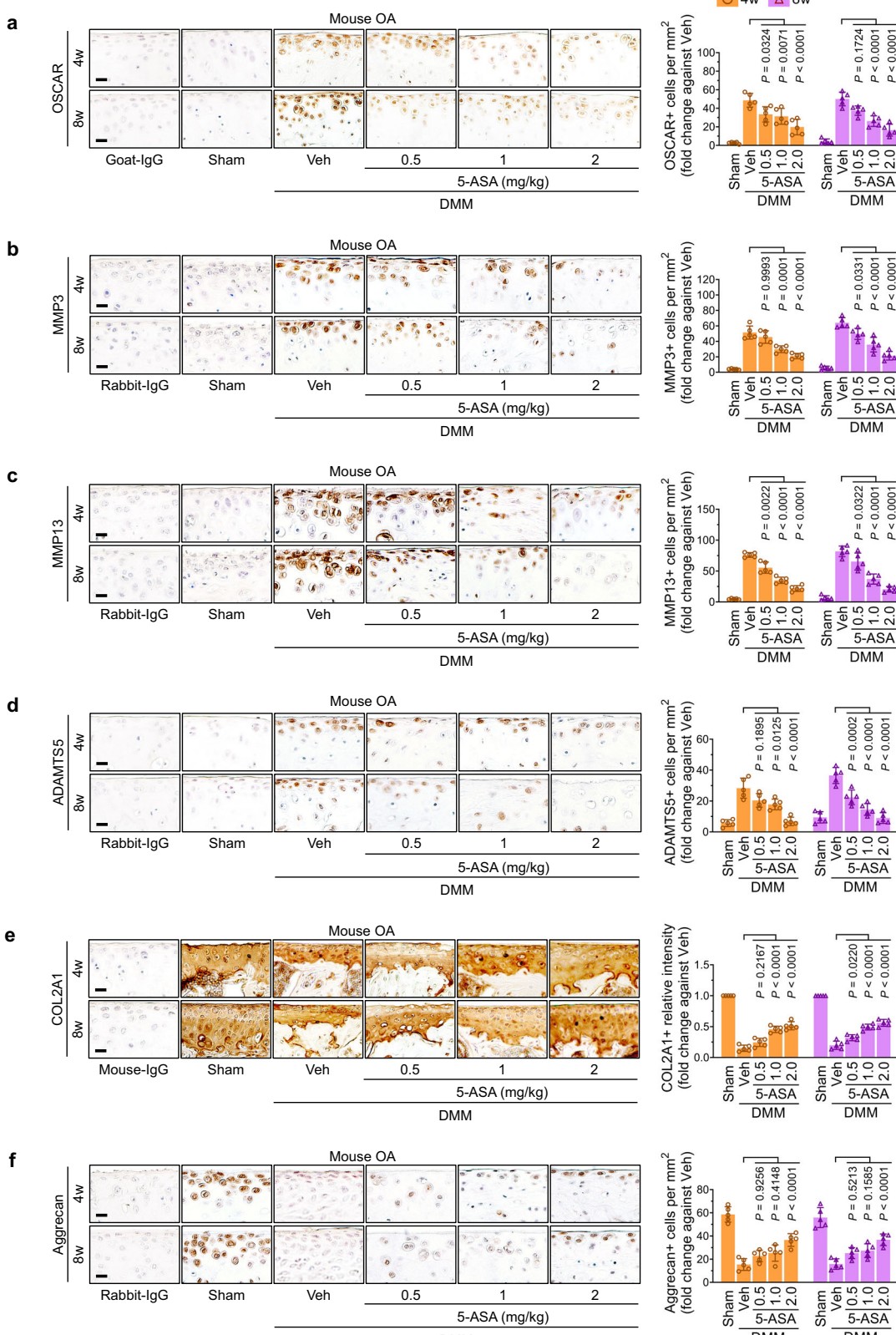

**Fig. 7 | 5-ASA reduces catabolism and enhances anabolism in OA mouse joint tissues. a–f** C57BL/6 J mice were subjected to sham operation or DMM surgery, IA-injected with vehicle (Veh) or 5-ASA (0.5, 1, and 2 mg/kg) for 8 weeks as shown in Fig. 5a, and sacrificed 9 weeks after the operation (*n* = 10 mice per each group). The joint tissues were subjected to immunohistochemistry for OSCAR (**a**), MMP3 (**b**),

MMP13 (**c**), ADAMTS5 (**d**), ACAN (**e**), and COL2A1 (**f**) and the protein expression levels were quantified. Scale bar, 25 μm. The data are shown as mean ± s.e.m. (*n* = 5 for each group; nonparametric test). *P*-values were obtained by two-way ANOVA was performed followed by Sidak's multiple comparisons test. Source data are provided as Source Data files.

experiment, OSCAR was knocked down with weekly IA injections of Ad-shOSCAR starting 1 week after DMM surgery and ending 1 week before sacrifice. In other experiments, the mice were IA injected with 0.5, 1, or 2 mg/kg 5-ASA or 2 mg/kg vehicle once weekly, either 8 times starting 1 week after surgery (i.e., treatment over the whole OA period) or 4 times starting 5 weeks after surgery (i.e., treatment for the last half of the OA period). In both regimens, the last 5-ASA injection occurred 1 week before sacrifice at week 9.

## Articular chondrocyte isolation and adenovirus-mediated OSCAR overexpression
Murine articular chondrocytes were isolated from the femoral condyles and tibial plateaus of 4–5-day-old ICR or C57BL/6 J WT mice by digestion with 0.2% collagenase type II. After culture for 48 h in Dulbecco's modified Eagle's medium (DMEM; HyClone, Logan, UT, USA) containing 10% fetal bovine serum (FBS), the cells were infected with the indicated MOIs of Ad-OSCAR or Ad-Control for 2 h and then cultured for an additional 24 h alone or with 5-ASA or pharmacological agents as described below. Normal human articular chondrocytes were isolated similarly from the normal healthy cartilage that was excised from a patient with OA who underwent total knee replacement surgery.

## Histological analysis of OA
The knee joints of mice were fixed in 10% formaldehyde at 4 °C for >24 h, decalcified in 0.5 M ethylenediaminetetraacetic acid in PBS (pH 7.4) for 2 weeks, embedded in paraffin, sliced into 5-μm sections, and stained with hematoxylin and eosin, 0.1% Safranin-O (s8884; Sigma-Aldrich, St Louis, MO, USA), and 0.05% Fast green FCF (f7258; Sigma-Aldrich, St Louis, MO, USA). Sclerosis and articular cartilage destruction were identified by safranin-O staining and measured with OsteoMeasureXP (OsteoMetrics, Inc., Atlanta, GA, USA), Image-pro plus (v4.5, Media Cybernetics, Inc., Rockville, USA), Adobe Photoshop (v9.0, San Jose, CA, USA), and an Olympus DP72 charge-coupled device camera (v2.1, Olympus Corporation, Tokyo, Japan). Articular cartilage destruction was scored by using OARSI grades (0–6), which is a standard OA-grading system[18,62] (Supplementary Data 4). The OARSI grades of the medial tibia were assessed by averaging the scores of three experienced investigators. The ratio of hyaline cartilage to calcified cartilage, osteophyte maturity, and synovitis[65] were also determined (Supplementary Data 5). Moreover, subchondral bone sclerosis was determined by measuring the SBP thickness.

## Generation of hOSCAR-Fc fusion protein and the Tyr166Ala, Tyr200Ala, and Ser166Ala mutants
293 F cells were transfected with a pVITRO1-Fc clone that expressed hOSCAR-Fc, namely, the extracellular domain (amino acids 19–233) of human OSCAR fused to the Fc region of human IgG1. The secreted purified protein was then loaded into a Thermo Scientific™ Pierce™ Protein G-Sepharose bead column (Thermo Fisher Scientific, Waltham, MA, USA) and eluted with elution buffer (100 mM glycine, pH 2.0, and 1 M Tris-Cl, pH 7.0; Duchefa Biochemie, 2003 RV Haarlem, the Netherlands) to immediately neutralize the protein. The tubes containing high concentrations of protein were collected, extensively dialyzed against PBS, and kept frozen at −80 °C. The Tyr166Ala, Tyr200Ala, and Ser166Ala mutants of hOSCAR-Fc were generated by a PCR-based method.

## High-throughput screening of small compound libraries
In total, the libraries contained 3287 compounds. They consisted of 1316 natural compounds from Med Chem Express (MCE; Monmouth Junction, NJ, USA) and 1971 from the FDA-approved Drug library (ApexBio Technology; Houston, TX, USA). For primary screening, we established an ELISA-based high-throughput screening method using hOSCAR-Fc fusion protein and the OSCAR-binding triple-helical

collagen peptide (COL$^{pep}$) in the 96-well plate scale. COL$^{pep}$ (short-form sequence: GPC-(GPP)$_5$-GPOGPAGFO-(GPP)$_5$-GPC-amide; alignment anchor residues are underlined) was from the University of Cambridge (CAS No. 2260906-29-2; Trinity Ln, CB2 1TN, UK). Its purity was confirmed by high-performance liquid chromatography. GPP10, which is a collagen-mimetic peptide (GPC-(GPP)$_{10}$-GPC-amide; CAS No. 2260816-94-0; University of Cambridge), served as a negative control peptide. Both were dissolved in 0.01 M acetic acid. Screening was initiated by pre-coating 96-well plates with 2 μg/ml COL$^{pep}$ overnight at 4 °C and pre-incubating each library compound at concentrations ranging from 0.0001 to 50 μM for 1 h with 0.1 μg/ml hOSCAR-Fc fusion protein. The hOSCAR-Fc mixtures were then added to the COL$^{pep}$-bearing plates for 1 h. Horseradish peroxidase-conjugated goat anti-human IgG (Cat# LF-SA8014H; Ab Frontier, 1:3000 dilution) served as the secondary antibody. The optical density at 450 nm (OD 450 nm) was then determined. All compounds were tested five times.

## Verification testing of candidate molecules identified by screening
For verification, the COL$^{pep}$/hOSCAR-Fc-based ELISA described above was used with increasing concentrations of candidate molecules. Moreover, to determine whether the candidates could compete with collagen-II for binding to cell-surface OSCAR, murine chondrocytes were cultured for 48 h in plates on which 2 μg/ml COL$^{pep}$ had been immobilized, infected with Ad-OSCAR for 2 h, and then treated with each candidate molecule for 48 h. Alternatively, murine chondrocytes were not infected with Ad-OSCAR. OSCAR mRNA/protein was isolated and measured by qRT-PCR and Western blotting, respectively.

## RNA isolation and qRT-PCR
Total RNAs from primary chondrocytes or bone marrow-derived MSCs were isolated by using TRIzol (Invitrogen, Carlsbad, CA, USA). Total RNAs from murine and human knee joint tissues were isolated by using the RNA Mini Kit (Life Technologies, Carlsbad, CA, USA). The RNAs were reverse-transcribed to generate complementary DNA (cDNA) by using the Superscript cDNA synthesis kit (Invitrogen) according to the manufacturer's instructions. qRT-PCR was performed using the KAPA SYBR Green fast qPCR kit (Kapa Biosystems, Inc., Wilmington, MA, USA) on a Step One Plus RT-PCR machine (Applied Biosystems, Foster City, CA, USA). The samples were tested in triplicate and the data were normalized by using β-actin as a housekeeping gene. Supplementary Data 6 provides the full list of primers.

## Primary culture of articular chondrocytes with COL$^{pep}$, IL-1β, 5-ASA, and/or rosiglitazone
The cultured chondrocytes were normal human or murine chondrocytes, or murine chondrocytes that were infected with Ad-OSCAR, Ad-Control, Ad-PPARγ, or Ad-PTGS2 at the indicated MOIs for 2 h. The chondrocytes were cultured in DMEM containing 10% FBS in the indicated conditions for 2 days before being infected with adenovirus or adding COL$^{pep}$ or other treatments. COL$^{pep}$ treatment involved culture for 48 h on dishes that were pre-coated with 2 μg/ml COL$^{pep}$ or GPP10 control peptide. IL-1β (10 ng/ml) treatment was conducted for the indicated time. The cells were treated with the indicated concentration of 5-ASA (or 10 μM 5-ASA) or 5-ASA vehicle (DMSO) for 24–48 h. Rosiglitazone (10 μM; R2408, Sigma Aldrich) treatment was conducted for 24–48 h.

## Western blotting analysis
Articular chondrocytes were lysed with lysis buffer (50 mM Tris-HCl, pH 8.0, 150 mM NaCl, 0.5% deoxycholate acid, and 1% NP-40) containing protease and phosphatase inhibitors. Antibodies against the following proteins were used for Western blotting analysis: OSCAR (Cat# PA5-47171; Thermo Fisher Scientific; 1:1000 dilution), MMP3 (Cat# Ab53015; Abcam; 1:1000 dilution), MMP13 (Cat# Ab39012; Abcam; 1:1000 dilution), ADAMTS5 (Cat# Ab41037; Abcam; 1:1000

dilution), COL2a1 (Cat# sc52658; Santa Cruz Biotechnology; 1:1000 dilution), ACAN (Cat# Ab3778; Abcam; 1:1000 dilution), SOX9 (Cat# 82630 s; Cell Signaling Technology; 1:1000 dilution), p-Syk (Cat# 2711 s; Cell Signaling Technology; 1:1000 dilution), Syk (Cat# 13198; Cell Signaling Technology; 1:1000 dilution), p-PLCγ2 (Cat# 3874 S; Cell Signaling Technology; 1:1000 dilution), PLCγ2 (Cat# sc407; Santa Cruz Biotechnology; 1:1000 dilution), iNOS (Cat# Ab178945; Abcam; 1:1000 dilution), COX-2 (Cat# 12282; Cell Signaling Technology; 1:1000 dilution), LOX-5 (Cat# 3289; Cell Signaling Technology; 1:1000 dilution). p-p38 (Cat# 9211 L, Cell Signaling Technology, 1:1000 dilution), p38 (Cat# 9212 L, Cell Signaling Technology, 1:1000 dilution), p-JNK1/2 (Cat# 9251 S, Cell Signaling Technology, 1:1000 dilution), JNK1/2 (Cat# 9252 S, Cell Signaling Technology, 1:1000 dilution), p-ERK (Cat# 9101 L, Cell Signaling Technology, 1:1000 dilution), ERK (Cat# 9102 S, Cell Signaling Technology, 1:1000 dilution), p-IκBα (Cat# 9246 S, Cell Signaling Technology, 1:1000 dilution), IκBα (Cat# 9242 S, Cell Signaling Technology, 1:1000 dilution), p-p65 (Cat# 3031 S, Cell Signaling Technology, 1:1000 dilution), p65 (Cat# 8242 S, Cell Signaling Technology, 1:1000 dilution). p-Akt (Cat# 9271, Cell Signaling Technology, 1:1000 dilution) and Akt (Cat# 9272, Cell Signaling Technology, 1:1000 dilution). Antibodies against β-actin (Cat# sc47778; Santa Cruz Biotechnology; 1:1000 dilution) and GAPDH (Cat# sc32233; Santa Cruz Biotechnology; 1:1000 dilution) served as loading controls.

### Induced-fit docking
All the docking and scoring calculations were performed using the Schrödinger software suite (Maestro, version 11.8.012). The SDF file of 5-ASA was acquired from the PubChem database. The file was imported into Maestro and arranged for docking using Ligand Preparation. The atomic coordinates of the crystal structure of OSCAR (Protein Data Bank; PDB ID: 5CJB) were saved from the PDB and prepared by eliminating all solvents, adding hydrogens, and minimal minimization using Protein Preparation Wizard. An ionizer was used to produce an ionized state of the three selected candidate compounds at the target pH of $7.0 \pm 2.0$. The input for IFD was the prepared low-energy ligand forms. The IFD protocol was processed on the graphical user interface. Namely, Maestro is linked with the Schrödinger software. Receptor sampling and refinement were conducted for residues within 5.0 Å of each ligand for each ligand–protein complex. IFD prime energy-minimizing with side-chain sampling, prediction module, and the backbone of OSCAR was conducted. The entire induced-fit receptor conformations generated with 5-ASA were scored by combining Prime and Glide Score scoring functions.

### RNA-seq analysis to identify OSCAR and 5-ASA downstream targets
Chondrocytes cultured for 2 days were infected with 800 MOI Ad-OSCAR for 2 h. They were then cultured with or without 5-ASA (10 µM) for an additional 24 h. As a control, the chondrocytes were injected with Ad-Control. All cultures were conducted in plates coated with COL$^{pep}$. IL-1β was not present in these cultures. Total RNA was extracted and sequencing libraries were prepared according to manufacturer instructions (TruSeq Stranded mRNA Library Prep Kit; Illumina, San Diego, CA, USA). Thus, paired-end sequencing was performed using an Illumina NovaSeq 6000 system according to the provided protocols for 2 × 100 sequencing. The sequencing quality of the FASTQ raw files was assessed by using FastQC. The RNA sequencing reads were trimmed by BBDuk and aligned to the GRCm38 genome reference by using STAR v2.7.1a. Raw read counts were obtained by RSEM v.1.3.1. The raw and processed RNA-seq data are available from the Gene Expression Omnibus (GEO) database (accession number GSE207056). The DEGs for OSCAR and 5-ASA were determined by comparing the Ad-Control chondrocytes to the untreated Ad-OSCAR-infected chondrocytes and the 5-ASA-treated Ad-OSCAR-infected chondrocytes, respectively. The differential expression analysis was performed by using DESeq2, and

the Benjamini-Hochberg method[66] was used for multiple test correction. Since OSCAR and 5-ASA have different ranges of gene expression regulation, we used alternative cut-off criteria to select the DEGs, namely, |log2FC| >1.5 and FDR < 0.05 for OSCAR and |FC| >1.5 and p-value < 0.05 for 5-ASA. The annotated human genes were mapped to mouse genes by using MGI orthology information. Enrichment factor (EF) was measured by calculating the overlap score ratio of actual value against expected value and significance was measured by using the hypergeometric test. Network analysis was conducted, with the pathways being grouped with K-means clustering. The PPI network was collected from the STRING database v.11.0[67]. Networks were visualized by Cytoscape v.3.8 software[68].

### CMap analysis
Potentially important 5-ASA targets were also identified by using the CMap dataset (L1000), which provides the transcriptome profiles after treatment with 20413 small molecules[69]. We screened this dataset for drugs that mimic 5-ASA by calculating the Jaccard index between the FlipDEGs and the 100 most up/downregulated genes in each drug profile. Only profiles with high-quality and FDA-approved drugs were used. Since many drugs had multiple profiles due to the use of different doses, harvesting times, and cell lines, a hypergeometric test was conducted with the top 10% of all Jaccard index scores. The target scores reflect the enrichment factor of drug–target interactions in the screened 5-ASA-mimicking drugs compared to the others. The Drug-Target interactions were collected from seven public databases, namely, ChEMBL, DGIdb, DrugBank, IUPHAR, KEGG, PharmGKB, and DrugCentral.

### Immunofluorescence analysis
Normal murine chondrocytes incubated with COL$^{pep}$, IL-1β, and/or 5-ASA were subjected to immunofluorescence staining of COX-2 (Cat# 12282; Cell Signaling Technology; 1:1000 dilution) and DAPI staining (D9542; Sigma Aldrich; 1:1000 dilution) and COX-2 fluorescence intensity was measured by Zeiss LSM 880 with Airyscan.

### ELISA for PGE2 and LTB4
Normal murine chondrocytes incubated with COL$^{pep}$, IL-1β, and/or 5-ASA were subjected to measure ELISA optical density. We used ELISA kits for PGE2 (R&D systems, #KGE004B) and LTB4 (R&D systems, #KGE006B) according to the manufacturer's instructions.

### Generation of bone marrow-derived MSCs, chondrogenesis, and 5-ASA treatment
Primary bone marrow mesenchymal stem cells were isolated from the tibias and femurs of 4–5-week-old male C57BL/6 J mice as previously described[70]. The cells were then incubated for 3 weeks in a chondrogenic medium (Hyclone, α-MEM containing 10 µg/ml insulin, 10 µg/ml transferrin, and 6 µg/ml sodium selenite) with or without 5-ASA (10 µM).

### Alcian-Blue staining of cartilage glycosaminoglycans and chondrogenic MSCs
Human knee joint sections were obtained as described above and bone marrow-derived MSCs cultured in the chondrogenic medium were stained with Alcian Blue (B8438; Sigma Aldrich) and photographed with Olympus DP72 charge-coupled device camera (v2.1, Olympus Corporation, Tokyo, Japan). The Alcian-Blue activity in the dishes was also quantified by measuring the absorbance at 630 nm.

### Immunohistochemistry
Human knee joint sections obtained as described above were incubated with primary anti-OSCAR antibodies (Cat# SC34235, Santa Cruz

Biotechnology, Inc., Dallas, TX, USA and Biorbyt, LLC, St Louis, USA, 1:200 dilution) at 4 °C overnight, followed by use of a 3,3′-diaminobenzidine peroxidase (DAB; horseradish peroxidase) substrate detection kit (Vector Laboratories, Inc., Burlingame, CA, USA) and counterstaining with hematoxylin. Immunohistochemistry was also conducted with antibodies against MMP3 (Cat# Ab53015; Abcam, Cambridge, MA, USA; 1:50 dilution), MMP13 (Cat# Ab51072; Abcam; 1:25 dilution), Aggrecan (Cat# Ab3778; Abcam; 1:100 dilution), COL2A1 (Cat# MAB8887; Sigma-Aldrich, St Louis, MO, USA; 1:50 dilution), ADAMTS5 (Cat# GTX100332; Genetex, Irvine, CA, USA; 1:200 dilution), SOX9 (Cat# ab185230; Abcam; 1:50 dilution), and PPARγ (Cat# ab59256; Abcam; 1:25 dilution).

### Pellet culture of chondrocytes and treatment with 5-ASA

To isolate bone marrow-derived mesenchymal stem cells (BM-MSCs) from 3- or 4-week-old mice, the following procedure was employed[71]. Firstly, the mice were euthanized, and their bodies were disinfected using ethanol. Subsequently, the limbs were carefully dissected, and the skin, muscles, ligaments, and tendons were meticulously removed. The tibia and femur bones were then thoroughly cleansed and transferred to a culture dish containing α-MEM medium, where they were cut into 4–5 pieces. Following a 2-day culture period, the bone pieces were discarded, and the remaining bone marrow was cultured for an additional 5 days. During this time, fusiform-shaped cells emerged by the third day, achieving a confluence level of 70–90% within the subsequent 2 days. The obtained BM-MSCs were further cultured in pellet form within a centrifuge tube, utilizing a cartilage-specific medium. After initiating the pellet culture, 2 μg/ml of COL$^{pep}$ was introduced into the medium on the following day. This medium was subjected to treatment with or without 5-ASA. The pellet culture was sustained for a duration of 3 weeks, with regular changes of the culture medium every 2–3 days.

### Human chondrocyte spheroid 3D culture

Human chondrocytes were subjected to spheroid culture in a 3D microenvironment with or without 5-ASA. A kit was used for this (Catalog No. SP3D-4650; Sciencell, City, Country). COL$^{pep}$ was added to the kit medium at 2 μg/ml at the start of the culture. After 1 week, the diameter of the spheroids was measured with a micro-ruler.

### Micro-CT analysis

The right and left hind limbs of 10-week-old male mice that had undergone DMM surgery or sham surgery 10 weeks previously were fixed in 10% formaldehyde, and the femurs and tibias were scanned by in vivo micro-CT (Skyscan1176, Bruker microCT, Kontich, Belgium). Supplementary Data 7 shows the micro-CT scanning parameters and reconstruction parameters used to acquire high-quality images. The raw micro-CT data were translated into 2-dimensional cross-sectional gray-scale image slices by using Nrecon (Brucker micro-CT, Kontich, ver.1.6.9.3, Belgium), after which the following structural variables of the trabecular and cortical bones were measured by CT Analyzer (CT-AN ver.1.10.9.0, Brucker, Belgium): the bone volume fraction (BV/TV, %), which is related to the shape of trabeculae. To assess the changes in the tibial SBP and subchondral trabeculae, both the SBP and remaining subchondral trabecular bone regions were manually segmented from the tibia, after which the BV was quantified.

### Statistical analyses

The sample size for each experiment was not predetermined. To analyze differences between 2 and >2 groups, students' two-tailed $t$-test, one-way analysis of variance (ANOVA), or two-way ANOVA were conducted, respectively. If ANOVAs were significant, pairwise multiple comparisons were conducted. Data based on the comparison of two samples with a variable measured using an ordinal grading system were analyzed using Sidak's multiple comparisons test. Data based on ordinal grading systems were analyzed using the non-parametric Mann–Whitney $U$ test. Each $n$ indicates the number of biologically independent samples, mice per group, or human specimens. The sample size for each experiment was not predetermined. The $P$ values are indicated in the figures or in Source Data, and the error bars represent s.e.m. for parametric data and the calculated 95% CIs for nonparametric data. Except where stated, the experiments were not randomized and the investigators were not blinded to allocation during experiments and outcome assessment. Multiple comparisons were performed using Tukey's test, Sidak's test, or Dunn's test with $P$ values set at <0.0001. All graphs and statistical analyses were made by using GraphPad Prism (v8.4.3, San Diego, CA, USA). $P$-values are indicated in the figures. The error bars in the figures show S.E.M. or 95% confidence intervals. All data were collected from at least three independent experiments.

### Reporting summary

Further information on research design is available in the Nature Portfolio Reporting Summary linked to this article.

## Data availability

The data that support the findings of this study are available within the article and its Supplementary Information files. Source data are provided in this paper. RNA-seq data of the Ad-OSCAR-infected and 5-ASA-treated chondrocytes have been deposited in the Gene Expression Omnibus (GEO) under accession number GSE207056. The CMap and transcription factor datasets utilized in this study are available at https://clue.io/ and https://maayanlab.cloud/Harmonizome/, respectively. The following figures have associated raw data: Figs. 1b, f, 1i, 3b, d, e, g, 4a–c,e–i, 5c, e, h, 6c, 7a–f; Supplementary Figs. 1c, 2a–c, g, 3a–c, f, h, 5a–g, 6b, d, 7b. For gel Source data, see Supplementary Figure 9. Source data are provided in this paper.

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

## Acknowledgements

The schematic illustration in the figure was created with BioRender.com. This work was supported by grants from the National Research Foundation of Korea (2021R1A2C3003675 and RS-2023-00217798 to S.Y.L.; 2022R1l1A1A01054308 to J.K.) and by the Korea Basic Science Institute National Research Facilities & Equipment Center grant (2019R1A6C1010020). We would like to thank Yongwon Choi (University of Pennsylvania, USA) for helpful discussions and suggestions.

## Author contributions
J.K. performed the experiments, analyzed the data, and co-wrote the paper. J.K., J.S., M.G., G.K., S.Y., S.K., D.H.S., and H.S. performed the in vitro and in vivo experiments. G.R. and W.K. analyzed the RNA-Seq data. H.L. and H.S.K. performed and analyzed the Micro CT data. J.Y.L. interpreted the data and provided scientific discussion. M.C.P. evaluated the human samples. S.Y.L. designed the study, analyzed and interpreted the data, and co-wrote the paper.

## Competing interests
The authors declare no competing interests.
