## [Peer Review File · Nature Communications]

REVIEWER COMMENTS

Reviewer #1 (Remarks to the Author):

This manuscript identifies 5-aminosalicylic acid (5-ASA) as a inhibitor of osteoclast-associated receptor (OSCAR). Focusing on the compound discovery and docking studies, an ELISA-based biochemical reconstitution assay was used to screen small molecules that could interfere/inhibit binding of a collagen II consensus peptide called "COLpep". Of 3 molecules shown to be active in this assay, only 5-ASA was able to inhibit OSCAR activity in cells. To determine the structural basis for inhibition, the authors docked 5-ASA onto OSCAR and described three residues (Y166, Y200, and S211) that may be important for the interaction. Mutation of these residues abolished the inhibitor effect of 5-ASA in the ELISA assay. These data appear to be sound. The mutational data provides support for the docked 5-ASA conformation. One recommendation: to the manuscript text would make the description of these data more informative for readers, it would be useful to describe what type of interactions occur between the OSCAR residues and 5-ASA (e.g., h-bonds, pi-stacking, etc.) to understand how the mutants may perturb these interactions.

Reviewer #2 (Remarks to the Author):

This is an interesting manuscript in which the author identify a pathogenic role of OSCAR in osteoarthritis with gain and loss of function studies, then, using a small molecule screen, they identify 5-ASA as an approved molecule which inhibits the binding of OSCAR with its ligand collagen type II and inhibits its pathogenic effects. The authors investigate molecular events downstream of OSCAR which are reverted by 5-ASA and convincingly prove the efficacy of 5-ASA in an instability-induced osteoarthritis model in mice.

The manuscript is timely, with osteoarthritis being a leading cause of disability worldwide so far without disease-modifying therapies, there are a lot of well-designed experiments and the experimentation is robust.

The main weaknesses are in order of priority: clarity must be made regarding the in-vivo models used and possibly the scoring should be repeated. There are some sweeping claims of causality where only association is shown which need toning down as associations or, ideally, functional experiments to prove causality, and statistical aspects to be corrected.

In detail:

- In figure 1E, I only see lots of safranin O staining which would be an OARSI score of 0.5, and yet in figure 1F the average of the treatment group injected with OSCAR adenovirus has a mean of 5 which would be loss of over half of the articular cartilage. Therefore either the images in figure 1e are not representative, or the A.OARSI score in figure 1F is not correct.

- The same applies to figures 1H-I and supplementary fig. 1. The images show mostly loss of safranin O staining (OARSI score 0.5) and superficial irregularity (OARSI score 1), but in graphs the scores are much higher. It is also not clear what scores are shown in the graph. Typically, the score from multiple sections within the weight bearing area of the cartilage are averaged for each bone and each compartment (max score 6), and shown separately and/or as a summer score (max score 24 if both medial and lateral compartment are included, or 12 if only the medial compartment is scored). Here the score is out of 6, therefore one assumes that this is the average, but of what? Of the medial tibia? The medial femur? All four components?

- Sampling is also a serious problem: in many sections, the meniscus almost completely separates the femur from the tibia, indicating that we are either in the most anterior or most posterior part of the joint, cutting through the anterior or posterior horns of the meniscus. The problem with that is that these portions are not weight bearing. the scores in the weight-bearing part are usually much higher. A rigorous sampling protocol is paramount for reliable scoring

Suppl fig. 2g: ID10, ID50 and ID90 should be calculated. By eye, it looks like 5-ASA is the worst of the three candidates.

- Extended fig. 2J, the text claims that it would show that “5- ASA displaced COLpep bound to OSCAR on chondrocytes”, however, mRNA levels of OSCAR are the readout, therefore, in my mind, the figure shows that 5-ASA reduces the expression of OSCAR following adenoviral expression, at transcriptional or post-transcriptional level. Similarly, in supp. 2k, it is OSCAR protein expression to drop after 5-ASA treatment. I am not clear how this has to do with the binding, at least directly. This is true also in Fig. 5E. I have the impression that yes, 5-ASA displaces collagen from OSCAR, but, possibly as a secondary effect, also downregulates OSCAR at transcriptional level.

- Extended figure 2j is not referred to in the text

- Fig 1h-i. This is, in terms of relevance, the cardinal figures of the paper. Yet, while we see in the graphs OARSI scores of 5 out of 6, in the images we only see loss of Safranin O staining (OARSI=0.5). Although

loss of proteoglycans, stained by safranin O, is a feature of osteoarthritis, without evidence of actual structural damage (erosions, clefts) is insufficient to claim osteoarthritis (PMID: 22777888) as such loss is usually reversible, unless associated with loss of collagen (see for instance PMID: 22777888; PMID: 18513402 and <https://arthritis-research.biomedcentral.com/articles/10.1186/ar2434>) and the simple loss of proteoglycans is insufficient. In addition, I do not see osteophytes or substantial bone sclerosis as would be evident, for instance, after joint destabilization. In summary, my impression, looking at the panels, is that OSCAR overexpression induces a marked loss of proteoglycans without structural damage to the cartilage and 5-ASA reverts it, which may be contributed by the well-known anti-inflammatory effect of 5-ASA.

- “Notably, 5-ASA is safe”. This statement was supported by rather superficial observation. Proper GLP toxicity studies are necessary for this type of statement. The statement should be moderated.

- As the OARSI score is a semi-quantitative score, non parametric tests should be used to compare groups. Not ANOVA.

The list of DEG in fig 2 must be made available, together with the raw data which should be deposited in a suitable repository such as GEO at NCBI.

p.9 li195-208 are very speculative and not substantiated by hard data.

- “ key transcription factors (TFs) that drive 5-ASA-induced flipping of the Flip^{catabolic} and Flip^{anabolic} pathways in Fig. 2c (Fig. 2d, Extended Data Fig. 3b). One was SOX9...”) but these are the transcription factors the expression of which changed after OSCAR/5-ASA treatment, not those that drive 5-ASA-induced flipping. To make that statement one would have to prove that forced activation of such transcription factors simulate the effect of 5-ASA and that in their absence 5-ASA becomes ineffective. Also the statements thereafter are very speculative, based on expression data and pathway analysis only, without real experimental, functional evidence.

- The experiment where the experimentally identified 5-ASA-induced targets, when inputted in the C-Map identified 5-ASA and correlated essentially with themselves is poorly informative, except perhaps that these might be “universal” 5-ASA targets identified in different experimental setup.

- The entire part (Fig. 3 and part of fig. 4) in which it is argued that the prostaglandin pathway (COX) may mediate the effect of 5-ASA, in reality only confirms that it is downstream, not that it mediates 5-ASA effects. In order to do so, one should perform gain and loss of function of for instance COX2 and demonstrate that, for instance, overactivation of the pathway phenocopies the effects of 5-ASA and that 5-ASA will not work in the absence of COX2. I am particularly unconvinced by this section because COX-2 inhibitors (the common non-steroidal anti-inflammatory drugs) have been used for over a century in the treatment of pain in osteoarthritis, but have never shown any disease-modifying effect.

The same applies to PPAR γ . To make the point that PPAR γ mediates the effect of 5-ASA, the authors need to show at least that 5-ASA is not chondrogenic in the absence of PPAR γ (for instance using PPAR γ siRNA or CRISPR). Ideally, also that PPAR γ overexpression phenocopies the effect of 5-ASA

Fig 3g should be analyzed with a two-way ANOVA.

- In figure 4 the authors claim that they are addressing the issue of whether 5-ASA promotes cartilage regeneration, however they only look at extracellular matrix production and some anabolic marker genes. To claim cartilage regeneration they need to ideally use a validated in vivo assay specific for cartilage regeneration (several injury models have been published), or at least show that in an osteoarthritis model, if they have three arms, with one arm being injected with vehicle at 8 weeks and the other two arms, one with vehicle and the other with 5-ASA, are killed after 10 weeks, the arm killed at 10 weeks with 5-ASA injection has cartilage that is thicker than the control arm killed at 8 weeks.

- Figure 5: It is claimed that DMM (destabilization of the medial meniscus) surgery is done to these mice, however, the degree of cartilage destruction is way too high compared to results by other investigators with this model. The DMM is a very mild model of instability-induced osteoarthritis where the scores are usually between 0.5 and 3, with rare 4 and exceptional 5, even at 9 weeks. Here there are a lot of 5 and 6. When one goes to the methods, however, one reads that the medial meniscus was not just destabilized, but actually removed, and a ligament is also mentioned. Which one? At the very least here we are talking about a model of medial meniscectomy, and possibly with resection of an additional ligament, which explains the high scores. The authors refer to reference 1 for the surgery, but it is clearly wrong because it refers to "The individual and socioeconomic impact of osteoarthritis". The model has to be clear and referenced correctly as this is a crucial aspect of this manuscript. The same applies to the scoring. What was scored? Only the medial compartment? Only the tibia? Both tibia and femur and then averaged? Summed scores? How many sections per knee were scored? What was the sampling protocol (only the sections in the weight bearing portion of the knee should be scored, as the portions covered by the menisci are usually relatively preserved. Finally, if the entire meniscus was actually removed as said in the methods, how come we see it in the sections? Was there meniscus regeneration?

- In Fig 5 one cannot use ANOVA for the OARSI score which is a semi-quantitative score. A non-parametric test should be used instead.

In reporting the results of Fig. 6 the authors claim that "5-ASA treatment in OA protects against cartilage destruction by downregulating OSCAR and ECM-degrading enzymes" whereas a simple association is shown. I would suggest toning down the statement with 5-ASA treatment in OA protects against cartilage destruction and is associated with downregulation of OSCAR and ECM-degrading enzymes

- Last year, Li et al. PMID: 36381242 reported the anabolic role of 5-ASA in chondrocytes and osteoarthritis, albeit they did not make the link with OSCAR and without the in vivo experimentation of

this manuscript. Although this paper does not take away any of the novelty of this study, it still needs to be referenced and commented upon.

Minor points

The results contain many aspects of methods and introduction. This is particularly evident, for instance, in end of p. 5 and the first half of p. 6.

Purely as a suggestion, in patients with ulcerative colitis 5-ASA is given orally. While this is not required for this manuscript, I think that there is an important opportunity if it was also active systemically, because osteoarthritis, in clinical practice, usually involves multiple joints and because the drug is already approved for clinical purposes.

Reviewer #3 (Remarks to the Author):

This is a potentially very interesting paper showing convincingly that OSCAR, a receptor first described as driving osteoclast maturation, can be blocked by the repurposed drug 5-ASA. Generally the data are of high quality with several in depth analyses to attempt to understand the mechanism of action, which appears to be through PPAR γ . Importantly, this identifies a novel class of anti-inflammatory that could be easily repurposed to test in human OA.

These data fit very well with other publications showing that PPAR γ KO mice develop accelerated disease (cited by authors) and that retinoic acid signalling, acting in a PPAR γ -dependent manner, is able to suppress mechanically induced inflammatory signalling and modify disease in surgically induced murine OA (DOI: 10.1126/scitranslmed.abm4054). The latter paper is not cited by the authors but is important as it sheds some further light on their findings. For instance, this paper demonstrates that retinoic acid is the limiting factor in PPAR γ -RXR anti-inflammatory actions. It is therefore perhaps highly relevant that RXRs are also regulated by OSCAR/5-ASA, and explains why, in Figure 4C, the addition of 5-ASA is required for PPAR γ agonists to have their full function. Furthermore, in Zhu et al, MAPK signalling was not affected by the anti-inflammatory effect of RXR-PPAR γ , suggesting that the anti-inflammatory effect was at the level of gene transcription, not signalling. This is consistent with Extended data Figure 5. Taken together the data suggest that OSCAR is enhancing PPAR γ action by affecting PPAR γ levels as well as promoting RXR-dependent function and this leads to suppression of inflammation by transcriptional regulation. I would strongly recommend that the authors revise their manuscript taking into consideration these complementary findings.

I have some additional specific comments:

1. The emphasis on COX2 is based on correlative studies, not mechanistic ones and I think this should be down-played in the manuscript. COX1, COX2 and COX1/2 KO mice show no protection in DMM induced OA (<https://doi.org/10.1002/art.33324>), nor do COX 2 antagonists modify human or murine disease (OA or RA). I think these likely represent biomarkers of an inflammatory pathway rather than effector molecules in OA pathogenesis.

2. I would also down-play the distinct role that 5-ASA has on chondrogenesis/repair. Inflammatory activators (such as IL1, TNF etc) are known to promote degradation whilst also suppressing anabolism. The effect of 5-ASA may therefore be indirect through the reduction in inflammation. The effect sizes in these anabolism pathways are also modest.

3. Where they are over-expressing OSCAR in the joint by Ad, I think it would be more accurate to describe this as OA-like disease rather than OA. It is an artificial model that does not mirror natural OA, and it looks as though there is principally proteoglycan depletion rather than breakdown of the cartilage (interestingly this is what is seen in the SOX9-aggrecan KO mice, (doi: 10.1002/jbmr.1696.)).

4. Do the authors have histology at 5 weeks i.e. prior to treatment with late 5-ASA (Fig 6). This would help establish whether there is reversal of disease or whether treatment simply halts progression from that time point.

5. Could the authors provide the list of the 124 first hit compounds in their screen and which ones have already been studied in OA – this would be of interest to the readers.

6. There are several important discussion points that I think are missing. Firstly, the authors say very little about what they think is happening in bone. Are the same inflammatory pathways activated?? Is this collagen II peptide activated (presume not). Is the collagen peptide exclusive to type II collagen or is it also in other collagens (type I and VI for instance)? And is it exposed on the native collagen or only exposed after degradation? 5-ASA is given to patients with UC orally with bioavailability designed to be bowel selective. Are oral preparations available that have more systemic effects? Why does overexpression of OSCAR in vivo lead to OA-like disease in the absence of col II peptide trigger? Does this mean that OSCAR can signal ligand-independently when overexpressed?

Minor comments:

- There is a lot of result in the latter part of the introduction which could be removed. Similarly there is quite a lot of discussion in the results e.g. lines 217-231.
- Extended Figure 2 has an extra panel (k) with consequent shunting of the other figure panels – the text is therefore out of synch.
- Line 113 please avoid saying “IL1b ...plays key pathogenic roles in OA” (e.g in lines 112 and 331, 340) - there is very little direct evidence to support a role for IL1 in knee OA either in mice (in KO studies) or in human (therapeutic studies) despite many people trying to demonstrate this. Many other cytokines or stimuli (e.g. mechanical damage directly) activate the same pathways as IL1.
- Also, there is no direct evidence that MMP3 is pathogenic. Indeed, MMP3-/- mice are not protected from DMM induced OA (several groups have shown this). Suggest remove the two references to MMP3.

- In line 246-7, is this result referring to Figure 1f? In which case, please add this. Line 110 please add reference to the collagen peptide binding to OSCAR.

Reviewer #1 Comments:

This manuscript identifies 5-aminosalicylic acid (5-ASA) as an inhibitor of osteoclast-associated receptor (OSCAR). Focusing on the compound discovery and docking studies, an ELISA-based biochemical reconstitution assay was used to screen small molecules that could interfere/inhibit binding of a collagen II consensus peptide called "COL^{pep}". Of 3 molecules shown to be active in this assay, only 5-ASA was able to inhibit OSCAR activity in cells. To determine the structural basis for inhibition, the authors docked 5-ASA onto OSCAR and described three residues (Y166, Y200, and S211) that may be important for the interaction. Mutation of these residues abolished the inhibitor effect of 5-ASA in the ELISA assay. These data appear to be sound. The mutational data provides support for the docked 5-ASA conformation.

Major comments

Q1.

One recommendation: to the manuscript text would make the description of these data more informative for readers, it would be useful to describe what type of interactions occur between the OSCAR residues and 5-ASA (e.g., h-bonds, pi-stacking, etc.) to understand how the mutants may perturb these interactions.

Reply: We sincerely appreciate your diligent review and your invaluable suggestion. To address it, we added the following text:

P8 L168-176 **"Induced-fit docking (IFD) studies showed that Tyr166, Tyr200, and Ser211 in OSCAR were important for its binding to epigallocatechin-3-gallate, morin hydrate, and 5-ASA (Supplementary Fig. 3g). More specifically, the IFD analysis showed that hOSCAR and 5-ASA likely bound stably because (i) the 5-amino and carboxylate groups of 5-ASA form hydrogen bonds with Ser211 and Tyr166 in hOSCAR-Fc, respectively; and (ii) the aromatic ring of 5-ASA interacts with Tyr200 through π - π stacking. This was confirmed by mutating each residue in 5-ASA to alanine and conducting the COL^{pep}/hOSCAR-Fc ELISA: each substitution abolished 5-ASA binding to OSCAR (Supplementary Fig. 3h)."**

Supplementary Information; P6 L59-66 Supplementary Fig. 3 **"Induced-fit docking analysis to determine the key OSCAR residues that bind to 5-ASA. Three different binding site poses of 5-ASA with hOSCAR-Fc were predicted and shown. The locations of the Tyr166, Tyr200 and Ser211 residues are indicated (arrows). The residues comprising the binding cavity are depicted with a stick model inside an electrostatic transparent surface potential mapping on OSCAR (left). The interaction pose is also displayed with a ribbon model (right). Red colors represent for negative charges and blue for positive, respectively. 5-ASA is drawn with thick sticks."**

Reviewer #2 Comments:

This is an interesting manuscript in which the author identify a pathogenic role of OSCAR in osteoarthritis with gain and loss of function studies, then, using a small molecule screen, they identify 5-ASA as an approved molecule which inhibits the binding of OSCAR with its ligand collagen type II and inhibits its pathogenic effects. The authors investigate molecular events downstream of OSCAR which are reverted by 5-ASA and convincingly prove the efficacy of 5-ASA in an instability-induced osteoarthritis model in mice.

The manuscript is timely, with osteoarthritis being a leading cause of disability worldwide so far without disease-modifying therapies, there are a lot of well-designed experiments and the experimentation is robust.

The main weaknesses are in order of priority: clarity must be made regarding the in-vivo models used and possibly the scoring should be repeated. There are some sweeping claims of causality where only association is shown which need toning down as associations or, ideally, functional experiments to prove causality, and statistical aspects to be corrected.

Reply: We are very grateful for your thoughtful, meticulous, and detailed review of our manuscript, your acute insights into the OA field, and your very welcome suggestions about how to improve our paper.

Reviewer #2 In detail:**Q1.**

In figure 1E, I only see lots of safranin O staining which would be an OARSI score of 0.5, and yet in figure 1F the average of the treatment group injected with OSCAR adenovirus has a mean of 5 which would be loss of over half of the articular cartilage. Therefore either the images in figure 1e are not representative, or the A.OARSI score in figure 1F is not correct.

Q2.

The same applies to figures 1H-I and supplementary fig. 1. The images show mostly loss of safranin O staining (OARSI score 0.5) and superficial irregularity (OARSI score 1), but in graphs the scores are much higher. It is also not clear what scores are shown in the graph. Typically, the score from multiple sections within the weight bearing area of the cartilage are averaged for each bone and each compartment (max score 6), and shown separately and/or as a summer score (max score 24 if both medial and lateral compartment are included, or 12 if only the medial compartment is scored). Here the score is out of 6, therefore one assumes that this is the average, but of what? Of the medial tibia? The medial femur? All four components?

Q3.

Sampling is also a serious problem: in many sections, the meniscus almost completely separates the femur from the tibia, indicating that we are either in the most anterior or most posterior part of the joint, cutting through the anterior or posterior horns of the meniscus. The problem with that is that these portions are not weight bearing. the scores in the weight-bearing part are usually much higher. A rigorous sampling protocol is paramount for reliable scoring.

Reply to Q1-3: We reviewed Figure 1 and agree Figure 1E was not at all consistent with the quantitative data shown in Figure 1F. This was also true for Fig. 1H-I and Supplementary Fig. 1. We found that this reflected both the selection of unrepresentative and poorly focused Safranin O images and an inadequate quantification system.

- To address this, we:
 - (i) repeated the experiments
 - (ii) carefully selected the images that were most representative of the OARSI, HC:CC, osteophyte, and synovitis scores
 - (iii) conducted all analyses in a blinded fashion with a careful and strictly automated OARSI scoring system. In particular, the OARSI grades (which were of the medial tibia) were assessed by averaging the scores of three experienced investigators. This system is based on the literature (refs 18, 62–64, which were added to the manuscript) and is detailed in Supplementary Tables 4 and 5. Briefly, the OARSI scoring system is as follows:

- score 0= Normal
- score 0.5= Loss of Safranin-O without structural changes
- score 1= Small fibrillations without loss of cartilage
- score 2= Vertical clefts down to the layer immediately below the superficial layer and some loss of surface lamina
- score 3= Vertical clefts/erosion to the calcified cartilage extending to <25% of the articular surface
- score 4= Vertical clefts/erosion to the calcified cartilage extending to 25-50% of the articular surface
- score 5= Vertical clefts/erosion to the calcified cartilage extending to 50-75% of the articular surface
- score 6= Vertical clefts/erosion to the calcified cartilage extending to >75% of the articular surface

- The new data are shown in Figure 1, Source data Fig.1, and at the end of this reply and are described in P5 L98-L104; P39 L900-L902; P42 L1013-1019; P44 L1036-L1052

- The fact that the medial tibia was used for the OASRI grading was added to the Methods:

P27 L602-607 “**Articular cartilage destruction was scored by using OARSI grades (0–6), which is a standard OA-grading system^{18,63} (Supplementary Table 4). The OARSI grades of the medial tibia were assessed by averaging the scores of three experienced investigators. The ratio of hyaline cartilage to calcified cartilage, osteophyte maturity, and synovitis⁶⁴ were also determined (Supplementary Table 5). Moreover, subchondral bone sclerosis was determined by measuring the SBP thickness.**”

- References were added:

18. Glasson SS, Blanchet TJ, Morris EA. The surgical destabilization of the medial meniscus (DMM) model of osteoarthritis in the 129/SvEv mouse. *Osteoarthritis Cartilage* **15**, 1061-1069 (2007).

62. Gardiner MD, *et al.* Transcriptional analysis of micro-dissected articular cartilage in post-traumatic murine osteoarthritis. *Osteoarthritis Cartilage* **23**, 616-628 (2015).

63. Glasson SS, Chambers MG, Van Den Berg WB, Little CB. The OARSI histopathology initiative - recommendations for histological assessments of osteoarthritis in the mouse. *Osteoarthritis Cartilage* **18 Suppl 3**, S17-23 (2010).

64. Krenn V, *et al.* Synovitis score: discrimination between chronic low-grade and high-grade synovitis. *Histopathology* **49**, 358-364 (2006).

Figure 1 e-f, h-i

Q4.

Suppl fig. 2g: ID10, ID50 and ID90 should be calculated. By eye, it looks like 5-ASA is the worst of the three candidates.

Reply: This reflects the use of a low COL^{pep} concentration (1.5 µg/ml) that led to saturation. In other words, the low COL^{pep} concentration meant that only a fraction of the added hOSCAR-Fc would bind to COL^{pep} collagen, which made it difficult to quantitatively observe the inhibitory effects of the three experimental agents.

To address this, we increased the concentration of COL^{pep} to 2 µg/ml and then performed a new experiment in which all other variables were controlled (Supplementary Fig. 3a, b). This showed that 5-ASA had the lowest IC₅₀ value of the three agents. The following texts were adapted:

P7 L145-L147 “Dose-ELISAs confirmed that these molecules suppressed hOSCAR-Fc binding to COL^{pep} (Supplementary Fig. 3a) and 5-ASA had the lowest IC₁₀, IC₅₀ and IC₉₀ value (Supplementary Fig. 3b).”

Supplementary Fig. 3a, b

Q5.

Extended fig. 2J, the text claims that it would show that “5- ASA displaced COL^{pep} bound to OSCAR on chondrocytes”, however, mRNA levels of OSCAR are the readout, therefore, in my mind, the figure shows that 5-ASA reduces the expression of OSCAR following adenoviral expression, at transcriptional or post-transcriptional level. Similarly, in supp. 2k, it is OSCAR protein expression to drop after 5-ASA treatment. I am not clear how this has to do with the binding, at least directly. This is true also in Fig. 5E. I have the impression that yes, 5-ASA displaces collagen from OSCAR, but, possibly as a secondary effect, also downregulates OSCAR at transcriptional level.

Reply: Yes, we did not explain the readout assay sufficiently. Our previous study (Ref 15) showed that when chondrocytes are cultured with collagen fragments and IL-1 β , OSCAR expression rises. This response is inhibited when hOSCAR-Fc is present. Thus, this chondrocyte assay is a measure of the binding of COL^{pep} to OSCAR on the surface of chondrocytes. Our data indeed suggest that 5-ASA displaces collagen from OSCAR and as a secondary effect (the readout), OSCAR is downregulated at a transcriptional level.

To address this, we added the following texts:

P6 L129-132 **“These OSCAR-transcription responses reflect downstream signaling induced by COL^{pep}-bound OSCAR that triggers new OSCAR expression¹⁵. These responses served as readouts of the binding of chondrocyte-surface OSCAR to COL^{pep} in our study.”**

P7-8 L160-166 **“...Similarly, 5-ASA antagonized the high OSCAR-protein and mRNA expression in IL-1 β +COL^{pep}-treated Ad-OSCAR-infected chondrocytes (Supplementary Fig. 3e, f). It should be noted here that since our screening ELISAs showed that 5-ASA binds directly to OSCAR, 5-ASA probably modulates OSCAR expression in chondrocytes by displacing COL^{pep} from cell-surface OSCAR, thus eliminating the downstream signaling induced by COL^{pep}-bound OSCAR that stimulates *de novo* OSCAR expression.”**

Supplementary Information; P3 L22-23 Supplementary Fig. 2 **“Readouts of the binding of COL^{pep} to OSCAR on the chondrocyte cell surface. This binding event leads to upregulation of OSCAR transcription¹⁵.”**

Supplementary Information; P6 L49-56 Supplementary Fig. 3 **“c-e, In vitro experiments to determine whether the three candidates could compete with COL^{pep} for OSCAR on the surface of primary murine chondrocytes. The binding of COL^{pep} to OSCAR on chondrocytes promotes OSCAR expression¹⁴. c, Chondrocytes were infected for 2h with Ad-OSCAR (MOI 800) on COL^{pep}-coated or uncoated plates without IL-1 β , treated for 48h with each of the three candidates. d-f, Chondrocytes were cultured for 48h on COL^{pep}-coated or uncoated plates without or with IL-1 β , and then treated with 5-ASA for 48h (d), or infected for 2h with indicated MOI of Ad-OSCAR (200, 400 or 800 MOI) (e, f).”**

Q6.

Extended figure 2j is not referred to in the text.

Reply: There was an error in the manuscript. Extended Figure 2J has now been relabeled as Supplementary Fig. 3f.

Q7.

Fig 1h-i. This is, in terms of relevance, the cardinal figures of the paper. Yet, while we see in the graphs OARSI scores of 5 out of 6, in the images we only see loss of Safranin O staining (OARSI=0.5). Although loss of proteoglycans, stained by safranin O, is a feature of

osteoarthritis, without evidence of actual structural damage (erosions, clefts) is insufficient to claim osteoarthritis (PMID: 22777888) as such loss is usually reversible, unless associated with loss of collagen (see for instance PMID: 22777888; PMID: 18513402 and <https://arthritis-research.biomedcentral.com/articles/10.1186/ar2434>) and the simple loss of proteoglycans is insufficient. In addition, I do not see osteophytes or substantial bone sclerosis as would be evident, for instance, after joint destabilization. In summary, my impression, looking at the panels, is that OSCAR overexpression induces a marked loss of proteoglycans without structural damage to the cartilage and 5-ASA reverts it, which may be contributed by the well-known anti-inflammatory effect of 5-ASA.

Reply: This was also a point raised by another reviewer, and we agree very much with it. Indeed, 3 weeks after weekly IA injections of Ad-OSCAR, the joint damage is limited to loss of proteoglycans and synovitis. However, when we extend the Ad-OSCAR injections for another 5 weeks (i.e. 8 weeks of treatment), we do observe changes in the OARSI grade, subchondral bone plate thickness, and osteophyte formation. Thus, structural changes to the cartilage do eventually arise if the Ad-OSCAR injections are given long enough. We do agree, however, that we should be more cautious in our description of these changes. Therefore, on the suggestion of the other reviewer, we have made sure to refer to the joint changes in the Ad-OSCAR-injected mice as “OA-like” throughout the manuscript. We also made this point early in the Results:

P5 L97-106 “...we IA-injected murine knees with Ad-OSCAR (3 weekly injections; Fig. 1c). By the end of the third week, this induced *in vivo* chondrocyte-overexpression of OSCAR (Fig. 1d), damaged the glycosaminoglycans in the articular cartilage, and induced synovitis. Notably, there was no obvious cartilage loss or osteophyte development, and the thickening of the subchondral bone plate (SBP) that is suggestive of sclerosis was not observed (Fig. 1e-f). However, as will be detailed later, when mice were treated for 8 weeks with IA Ad-OSCAR injections, cartilage destruction, osteophyte formation, SBP thickening, and synovitis were observed (Fig. 1h-i). Nonetheless, it should be noted that the Ad-OSCAR-induced model involves OA-like disease rather than classical OA.”

Q8.

“Notably, 5-ASA is safe”. This statement was supported by rather superficial observation. Proper GLP toxicity studies are necessary for this type of statement. The statement should be moderated.

Reply: We agree and have moderated the statement as shown below. We also show below the results of H&E staining of cartilage tissue from mice treated with 5-ASA after DMM surgery to demonstrate that the rats show no morphological abnormalities in their organs, body length, body weight, and tibia length, even after 10 weeks of 5-ASA treatment, (Author response Fig. 1a, b). Nonetheless, we agree that more thorough safety analyses are needed, and thus have decided not to include these data in the manuscript.

P9 L192-194 “It should be noted that mice treated weekly with IA 5-ASA injections for 3 or 8 weeks were viable, normally sized, had normal lifespans, and lacked gross morphological or histological abnormalities.”

Author response Fig. 1. IA injection of 5-ASA shows no obvious abnormalities in the multiple tissues

a, b C57BL/6-J mice underwent either Sham or DMM surgery. The DMM-operated mice received IA injections once a week for 8 weeks with either a Veh or different doses of 5-ASA (0.5, 1, and 2 mg/kg) and were then sacrificed 9 weeks after the initial operation ($n = 10$ mice in each group). **a** H&E staining was performed on various tissues from both Sham mice and DMM-operated mice treated with either Veh or 5-ASA, as indicated. The scale bar in the images represents 50 μm . Representative images revealed no noticeable abnormalities in the thymus, heart, lung, liver, spleen, kidney, and testis of sham-operated mice in comparison to DMM-operated mice who received IA injections once a week for 8 weeks with either Veh or 5-ASA at doses of 0.5, 1, and 2 mg/kg. **b** Quantitative analysis of body length, body weight, and tibia length in male mice was conducted. The error bars in **b** represent the mean \pm s.e.m.

One-way ANOVA was performed, followed by Tukey's multiple comparison's test in (b). Notably, no statistically significant differences (ns, $P > 0.05$) were observed.

Q9.

As the OARSI score is a semi-quantitative score, non-parametric tests should be used to compare groups. Not ANOVA.

Reply: Thank you very much. To address this point and other such points, we checked all of our statistics. Subsequently, we conducted non-parametric Mann-Whitney U test or Kruskal-Wallis analyses on the OARSI grade, synovitis and osteophyte formation data. Since the HC/CC ratio and SBP thickness data are normally distributed, they were analyzed with Two-tailed t -tests. These changes have been incorporated in Figures 1, 5 and 6 and Supplementary Figures 1 and their figure legends.

e.g. P44 L1125-L1130 Figure 1 “**The OARSI grade, synovitis and osteophyte maturity data are shown as means \pm 95% confidence intervals (CI). Differences between groups were determined with Kruskal-Wallis test followed by Mann–Whitney U test. The HC/CC ratio and SBP thickness data were shown as means \pm s.e.m. Differences between groups were determined with two-tailed t -test. Exact P values can be found in the accompanying Source Data. Scale bars, 25 μm .**”

Q10.

The list of DEG in fig 2 must be made available, together with the raw data which should be deposited in a suitable repository such as GEO at NCBI.

Reply: To address this, we have added Supplementary Table 2, which lists all the DEGs. The raw data were already deposited in GEO (GSE207056), as indicated by the Data Availability statement (P36 L835) in the original manuscript.

Q11.

p.9 li195-208 are very speculative and not substantiated by hard data.

Reply: We agree and have revised the entire paragraph to remove weak statements that are insufficiently supported by the evidence. Relevant references that support our interpretations have also been added. The revised paragraph is as follows:

P10-11 L222-L237 “**Moreover, pathway analysis showed that known cartilage catabolism and anabolism pathways in OSCAR-overexpressing chondrocytes were altered by 5-ASA treatment. Specifically, the Flip^{catabolic} pathways included the inflammation and eicosanoid-related pathways (Fig. 2c). In particular, the Flip^{catabolic} genes in the eicosanoid-related pathway included prostaglandin-endoperoxide synthase 2 (*Ptgs2*, which encodes cyclooxygenase [COX]-2), arachidonate 5-lipoxygenase (*Alox5*, which encodes LOX-5), and their downstream genes (e.g. *Ptges*, *Alox5ap*, *Ltc4s*, *Ltb4r1*, and**

Ltb4r2) (Supplementary Fig. 4a). Moreover, the Flip^{anabolic} pathways included those that relate to collagen, NMDA-receptor & gap-junction trafficking, cholesterol-metabolism, and folate/amino-acid metabolism (Fig. 2d). Several of the latter are known to relate to cartilage-regeneration processes, including collagen production²² and glycine/serine metabolism, which is required for the biosynthesis of ECM collagen and glycoprotein²³. These transcriptomic changes together with the *in vitro* chondrocyte, induced docking, and Ad-OSCAR-injection data support the notion that 5-ASA may exert chondroprotective effects by altering the COL^{PEP}-stimulated signaling of OSCAR on chondrocytes.”

Q12.

“key transcription factors (TFs) that drive 5-ASA induced flipping of the Flip^{catabolic} and Flip^{anabolic} pathways in Fig. 2c (Fig. 2d, Extended Data Fig. 3b). One was SOX9...) but these are the transcription factors the expression of which changed after OSCAR/5-ASA treatment, not those that drive 5-ASA-induced flipping. To make that statement one would have to prove that forced activation of such transcription factors simulate the effect of 5-ASA and that in their absence 5-ASA becomes ineffective. Also, the statements thereafter are very speculative, based on expression data and pathway analysis only, without real experimental, functional evidence.

Reply: The TFs were those involved in 5-ASA-induced flipping but yes, the word “drive” was inappropriate here. Indeed, without further experiments, we cannot be sure that any of the TFs that we identified played a role in the 5-ASA-flipped pathways. We also agree that many of the statements were similarly speculative. To address this, we revised the entire paragraph as follows:

P11 L238-249 “To assess the changes further, we identified the 14 transcription factors (TFs) whose expression was most strongly altered by 5-ASA treatment (Fig. 2e) and then conducted network analysis (Fig. 2f). This suggested that PPAR γ -encoded PPARG may play an important role in the effect of 5-ASA on chondrocytes. It was downregulated by OSCAR overexpression but this was flipped by 5-ASA. This is consistent with studies showing that 5-ASA upregulates PPAR γ in epithelial cells²⁴. Other important TFs may be EP300, which is a co-activator of PPAR γ ²⁵; SOX9, which is a cartilage-anabolism marker in OA²⁶; SREBF1, which mediates cholesterol metabolism²⁷, which is one of the 5-ASA-regulated pathways (Fig. 2); ATF4, which promotes SREBF1 by inhibiting its degradation²⁸ and may upregulate collagen synthesis²⁹ and amino-acid metabolism³⁰; and RXRA, which heterodimerizes with PPAR γ and may thereby regulate lipid/cholesterol metabolism³¹.”

Q13.

The experiment where the experimentally identified 5-ASA-induced targets, when inputted in the C-Map identified 5-ASA and correlated essentially with themselves is poorly informative, except perhaps that these might be “universal” 5-ASA targets identified in different experimental setup.

Reply: We believe that we did not explain the purpose of this experiment sufficiently. The aim of this *in silico* analysis was to identify likely targets of 5-ASA by using the DEG profile that it induced to search for similar DEG profiles in CMAP, which describes the known targets of >20K CMAP drugs/compounds. Each drug typically has 10–30 targets. Such CMAP similarity analyses have been used extensively in the literature: indeed, there are >2,300 citations with CMAP v1 (Science 2006) and >1,100 citations for CMAP v2 (Cell 2017). Two examples of studies using CMap (Author Response References) are as follows:

Author Response References

1. Schenone, M., Dančik, V., Wagner, B. et al. Target identification and mechanism of action in chemical biology and drug discovery. *Nat Chem Biol* 9, 232–240 (2013). <https://doi.org/10.1038/nchembio.1199>
2. Wang K, Sun J, Zhou S, Wan C, Qin S, et al. (2013) Prediction of Drug-Target Interactions for Drug Repositioning Only Based on Genomic Expression Similarity. *PLOS Computational Biology* 9(11) <https://doi.org/10.1371/journal.pcbi.1003315>

To address this, we improved our explanation of this analysis and the results. The revised paragraph is as follows:

P11-12 L250-266 **“To further determine which FlipDEGs and FlipTFs could be particularly important 5-ASA targets in OA, we asked whether the 5-ASA-altered DEGs demonstrated similar expression patterns in response to other known drugs/compounds. For this, we conducted *in silico* analysis with Connectivity Map (CMap), a large dataset comprising the transcriptomes of >20K drugs/compounds that has been used extensively for drug repurposing and mode-of-action analyses³². This analysis indicated how closely the 5-ASA-induced Flip-DEG profile resembled the DEG profiles induced by each of the >20K drugs/compounds. We then took the top 10% most similar CMap profiles and listed the most enriched targets (see Methods for details). This revealed eight putative 5-ASA targets. Three have already been noted in the analyses above, namely, PTGS2/COX-2, RXRA, and PPARG. The other five predicted targets were ERBB2, TBXAS1, NR1H2, PPARD, and RARG (Fig. 2g): these were also in the Flip^{catabolic} and Flip^{anabolic} gene-sets (Supplementary Table 2) and two (PPARD and RARG) were in the 14 TFs that we found were most strongly altered by 5-ASA treatment (Fig. 2e). Moreover, Zhu et al.³³ showed that an RXR agonist can suppress OA and that this relies on activation of PPAR γ . In addition, like PTGS2/COX-2, TBXAS1 is also a key enzyme in the eicosanoid pathway (Supplementary Fig. 4a, b).”**

Q14.

The entire part (Fig. 3 and part of fig. 4) in which it is argued that the prostaglandin pathway (COX) may mediate the effect of 5-ASA, in reality only confirms that it is downstream, not that it mediates 5-ASA effects. In order to do so, one should perform gain and loss of function of for instance COX2 and demonstrate that, for instance, overactivation of the pathway phenocopies

the effects of 5-ASA and that 5-ASA will not work in the absence of COX2. I am particularly unconvinced by this section because COX-2 inhibitors (the common non-steroidal anti-inflammatory drugs) have been used for over a century in the treatment of pain in osteoarthritis, but have never shown any disease-modifying effect.

The same applies to PPAR γ . To make the point that PPAR γ mediates the effect of 5-ASA, the authors need to show at least that 5-ASA is not chondrogenic in the absence of PPAR γ (for instance using PPAR γ siRNA or CRISPR). Ideally, also that PPAR γ overexpression phenocopies the effect of 5-ASA.

Reply: Our reply is in two parts:

(Part 1) We agree that the general view is that targeting COX-2 does not associate with disease-modifying effects in OA. However, these drugs are delivered systemically, and a recent systematic review suggests that some COX-2 inhibitors could act as DMOADs if they are administered intra-articularly (Timur et al. 2020; ref 50). Moreover, another systematic review has found that even when delivered systemically, some COX-2 inhibitors (e.g. celecoxib) can modify the cartilage, synovium, and bone changes in OA (Nakata et al. 2018; ref 51). Therefore, we believe that the PPAR γ -COX-2-mediated anti-inflammatory activity induced by IA injections of 5-ASA could have the potential to exert chondroprotective effects. We added some text to the Discussion about this, as follows. Nonetheless, since we recognize that it is speculative that COX-2 inhibitors can have DMOAD activity, we also qualified the title of the paper and other texts:

Added text:

P21-22 L472-486 Discussion “...we found that 5-ASA competed with collagen for binding to OSCAR and its binding to OSCAR upregulated PPAR γ . This downregulated the eicosanoid pathway, including COX-2 and its generation of PGE2 (Supplementary Fig. 8). The human osteochondral explant study of Li et al.⁴⁸ also observed that culture of the explants with 5-ASA downregulated COX-2. It should be noted that while it is widely believed that systemic administration of COX-2 inhibitors (i.e. common nonsteroidal anti-inflammatory drugs) does not associate with chondroprotective effects in OA, a recent systematic review suggested that some COX-2 inhibitors could act as DMOADs if they are injected IA⁵⁰. Moreover, another recent systematic review found that even when injected systemically, some COX-2 inhibitors (e.g. celecoxib) can modify the cartilage, bone, and synovial disease in OA, and these effects are mediated by the regulation of prostaglandins and direct changes to the tissues⁵¹. Since our 5-ASA treatment in mice involves IA injections and is potentially a novel drug for OA, it is thus possible that COX-2 and LOX-5 could be effectors of OA. Alternatively, they are simply markers of the eicosanoid pathway.”

Altered texts:

P1 L1-2 Paper title “5-aminosalicylic acid suppresses osteoarthritis through the OSCAR-PPAR γ axis”

P12 L268-269 “PPAR γ signaling in chondrocytes may mediate the cartilage-protective

effects of 5-ASA”

P12-13 L282-286 “Since elucidating the molecular mechanisms by which 5-ASA protects cartilage could reveal therapeutic targets, we focused further on COX-2 and LOX-5 as either markers of the eicosanoid pathway or true targets. In either case, exploring the link between OSCAR, PPAR γ , COX-2, and LOX-5 could help illuminate the key role of PPAR γ in OA pathogenesis.”

The systematic reviews were also added as references:

P41 L1030-1036 References

50. Timur UT, *et al.* Chondroprotective Actions of Selective COX-2 Inhibitors In Vivo: A Systematic Review. *Int J Mol Sci* **21**, (2020).
51. Nakata K, *et al.* Disease-modifying effects of COX-2 selective inhibitors and non-selective NSAIDs in osteoarthritis: a systematic review. *Osteoarthritis Cartilage* **26**, 1263-1273 (2018).

(Part 2) We agree that we should show that the effects of 5-ASA on PPAR γ -COX-2 signaling in chondrocytes depend on the function of both COX-2 and PPAR γ . To address this, we conducted new experiments with four different adenoviruses, namely, Ad-Ptgs2, Ad-shPtgs2, Ad-Pparg, and Ad-shPparg. Ad-Ptgs2 and Ad-Pparg induce *Ptgs2*/COX-2 and PPAR γ overexpression, respectively while Ad-shPtgs2 and Ad-shPparg silence these genes. The results are presented in Supplementary Fig. 5 (see figures at the end of this reply); some other results were already shown previously (Fig. 4b, c). The results are as follows:

- When OSCAR was upregulated on primary chondrocytes with IL-1 β +COL^{pep}, 5-ASA reduced the high COX-2 (*Ptgs2*) mRNA levels induced by Ad-Ptgs2 infection (Supplementary Fig. 5a, b).
- Ad-Ptgs2 infection by itself elevated the mRNA of not only *Ptgs2* but also the Flip^{catabolic} DEGs *Mmp3*, *Mmp9*, *Mmp13*, and *Adamts5* (Supplementary Fig. 5c).
- Similarly, when OSCAR was upregulated on primary chondrocytes with IL-1 β +COL^{pep} and PPAR γ was overexpressed by Ad-Pparg, *Ptgs2*/COX-2 levels dropped markedly. This drop was even greater when 5-ASA was also present (Fig. 4b, c).
- IL-1 β +COL^{pep}-treated primary chondrocytes produced PGE2 but not when *Ptgs2*/COX-2 was silenced with Ad-shPtgs2. Adding 5-ASA did not increase this suppressive effect, which suggests that 5-ASA suppressed PGE2 release by suppressing *Ptgs2*/COX-2 (Supplementary Fig. 5d, e).
- Ad-Pparg infection by itself elevated the mRNA of not only PPAR γ but also the Flip^{anabolic} DEGs *RXRA*, *COL2a1*, *Acan*, and *SOX9* (the latter three are also known markers of chondrocyte anabolism) (Supplementary Fig. 5f).
- When bone marrow-derived MSCs were infected with Ad-shPparg and cultured with COL^{pep} in chondrogenic medium, 5-ASA could no longer increase cell expression of *COL2a1*, *Acan*, and *SOX9* (Supplementary Fig. 5g). This novel observation is important because it suggests that (i) PPAR γ can induce beneficial cartilage

anabolism (as well as decrease harmful joint inflammation) and (ii) 5-ASA can promote the differentiation of cartilage-anabolic chondrocytes from MSCs in a PPAR γ -dependent manner.

The following texts were added:

P13-14 L295-310 “We then confirmed that 5-ASA downregulated *Ptgs2*/COX-2 expression *via* OSCAR by infecting chondrocytes with Ad-Ptgs2 and then treating them with IL-1 β +COL^{pep} with or without 5-ASA: 5-ASA significantly reduced the overexpression of *Ptgs2*/COX-2. This was not observed when IL-1 β +COL^{pep} was not present which indicates that 5-ASA suppressed *Ptgs2*/COX-2 expression *via* OSCAR (Supplementary Fig. 5a, b). The gain-of-function experiments conducted with COX2 *via* Ad-Ptgs2 revealed that the overactivation of the eicosanoid pathway results in an opposing effect when compared to the impact of 5-ASA. The overexpression of *Ptgs2* exhibited a contrasting effect to that of 5-ASA concerning the expression of Flip^{catabolic} DEGs, including well-known markers of cartilage catabolism, such as *Mmp3*, *Mmp9*, *Mmp13*, and *Adamts5* mRNA, in chondrocytes (Supplementary Fig. 5c). The possibility that 5-ASA blocked COL^{pep}-induced OSCAR-mediated PGE2 production *via* *Ptgs2*/COX-2 was confirmed by infecting chondrocytes with Ad-sh*Ptgs2*, which silences *Ptgs2*/COX-2: when these cells were stimulated with COL^{pep}, they were unable to produce PGE2, as expected, and 5-ASA treatment had no effect on this (Supplementary Fig. 5d, e).”

P14-15 L321-333 “Next, we induced chondrocytes to overexpress PPAR γ with Ad-*Pparg* (Fig. 4b), cultured them with IL-1 β +COL^{pep} to upregulate OSCAR expression, treated them with 5-ASA, and measured *Ptgs2*/COX-2 expression. While IL-1 β +COL^{pep} increased *Ptgs2*/COX-2 expression as expected, overexpressing PPAR γ halved that effect, and this was further augmented when 5-ASA was also present (Fig. 4c). Moreover, overexpressing *Pparg* had the same effect as 5-ASA in terms of the Flip^{anabolic} DEGs (and known markers of cartilage anabolism^{7,8}) *Rxra*, *Col2a1*, *Acan*, and *Sox9* mRNA expression by the IL-1 β +COL^{pep}-treated chondrocytes: both treatments increased these mRNA levels (Supplementary Fig. 5f). These results together suggest that (i) when OSCAR expression in chondrocytes is increased by collagen binding, PPAR γ is downregulated, which upregulates *Ptgs2*/COX-2 expression and thereby elevates PGE2 secretion and inflammation; and (ii) 5-ASA treatment reverses this pro-inflammatory effect on the PPAR γ -COX-2-PGE2 axis.”

P16 L361-367 “Interestingly, if the bone marrow-derived MSCs were infected with Ad-sh*Pparg* before culture in chondrogenic medium containing COL^{pep}, 5-ASA could no longer increase cell expression of *Col2a1*, *Acan*, and *SOX9* (Supplementary Fig. 5g). This suggests that (i) PPAR γ can induce beneficial cartilage anabolism (as well as antagonize harmful eicosanoid pathway-mediated joint inflammation), and (ii) 5-ASA can promote the production of cartilage-anabolic chondrocytes in a PPAR γ -dependent manner.”

Supplementary Information P9-10, L94-116 Supplementary Fig. 5 legend.

Q15.

Fig 3g should be analyzed with a two-way ANOVA.

Reply: We did in fact apply a two-way ANOVA to Fig. 3g but erroneously reported it as a one-way ANOVA. This error has been corrected.

Q16.

In figure 4 the authors claim that they are addressing the issue of whether 5-ASA promotes cartilage regeneration, however they only look at extracellular matrix production and some anabolic marker genes. To claim cartilage regeneration they need to ideally use a validated in vivo assay specific for cartilage regeneration (several injury models have been published), or at least show that in an osteoarthritis model, if they have three arms, with one arm being injected with vehicle at 8 weeks and the other two arms, one with vehicle and the other with 5-ASA, are killed after 10 weeks, the arm killed at 10 weeks with 5-ASA injection has cartilage that is thicker than the control arm killed at 8 weeks.

Reply: We very much agree and greatly appreciate your suggestion. We therefore added our latest experimental data, which showed that mice that were treated with 5-ASA at weeks 8–11 and killed at week 12 had significantly thicker cartilage than mice that were treated with vehicle at 4–7 weeks and killed at week 8. Moreover, the OARSI scores (and other OA variables) of the 5-ASA-treated mice killed at week 12 were significantly lower than those of the week-8-killed mice. Note that we used a 4-week 5-ASA treatment period because we found that the most robust cartilage repair occurs when the total duration of 5-ASA treatment is at least 4 weeks. The new data are shown in Fig. 6. The following texts were added:

P2 L39-41 Abstract “Moreover, mice with DMM-induced OA that were treated with 5-ASA at weeks 8-11 and sacrificed at week 12 had thicker cartilage than untreated mice that were sacrificed at week 8.”

P4 L81-82 “Late injections at weeks 8-11 also led to thicker cartilage compared to untreated mice that were sacrificed at week 8.”

P18 L414-424 “We then showed that 5-ASA could even induce DMM surgery-damaged cartilage to recover when it was first started after the disease had progressed to severe degenerative OA, namely, 8 weeks after DMM (Fig. 6a). Hence, mice subjected to a 4-week treatment with 5-ASA from weeks 8-11 exhibited a decrease in several OA markers when contrasted with mice at the 12-week post-DMM surgery stage. In fact, the final OARSI grade of the 5-ASA-treated mice sacrificed at week 12 was much lower than that of untreated mice that were sacrificed at week 8. This was also true for the other OA indicators. Significantly, the treated mice that were killed at week 12 had thicker cartilage than the mice that were sacrificed at week 8 (Fig. 6b, c). These findings suggest that 5-ASA could not just arrest OA progression, it could reverse it. However, additional studies are needed to confirm this.”

P54 L1225-1234 Fig. 6 legend. “a Mice were subjected to sham operation or DMM surgery and then IA-injected once per week for the indicated 4-week prior to sacrifice with vehicle (Veh; G1 and G2) or 5-ASA (2 mg/kg; G3) and sacrificed 9 (G1) or 12 weeks (G2 and G3) after the operation ($n = 10$ mice per group). b Representative Safranin-O staining and fast green counter staining images. Scale bar, 25 μm . c Quantitation of the following OA variables: OARSI grade, osteophyte maturity, subchondral bone plate thickness, and synovitis. The OARSI grade, synovitis and osteophyte maturity data are shown as means \pm 95% confidence intervals (CI). Differences between groups were determined with Kruskal-Wallis test followed by Mann-Whitney U test. Means \pm s.e.m. with two-tailed t -test for SBP thickness. Exact P values can be found in the accompanying Source Data. Scale bars, 25 μm .”

Fig. 6

Q17.

Figure 5: It is claimed that DMM (destabilization of the medial meniscus) surgery is done to these mice, however, the degree of cartilage destruction is way too high compared to results by other investigators with this model. The DMM is a very mild model of instability-induced osteoarthritis where the scores are usually between 0.5 and 3, with rare 4 and exceptional 5, even at 9 weeks. Here there are a lot of 5 and 6. When one goes to the methods, however, one reads that the medial meniscus was not just destabilized, but actually removed, and a ligament is also mentioned. Which one? At the very least here we are talking about a model of medial meniscectomy, and possibly with resection of an additional ligament, which explains the high scores. The authors refer to reference 1 for the surgery, but it is clearly wrong because it refers to “The individual and socioeconomic impact of osteoarthritis”. The model has to be clear and referenced correctly as this is a crucial aspect of this manuscript. The same applies to the scoring. What was scored? Only the medial compartment? Only the tibia? Both tibia and femur and then averaged? Summed scores? How many sections per knee were scored? What was the sampling protocol (only the sections in the weight bearing portion of the knee should be scored, as the portions covered by the menisci are usually relatively preserved. Finally, if the entire meniscus was actually removed as said in the methods, how come we see it in the sections? Was there meniscus regeneration?

Reply: The reference that we gave for the DMM model was Glasson et al. (2007), which is titled "Surgical destabilization of a medial meniscus (DMM) model of osteoarthritis in 129/SvEv mice" and was originally *Online method reference #1* (the first of the *manuscript body* references was entitled “The individual and socioeconomic impact of osteoarthritis”, which indeed is not relevant to DMM). To avoid confusion between paper body references and Online method references, we modified the manuscript so that the 2007 Glasson et.al. paper and their follow-up paper in 2010 are now cited as References 18 and 63, respectively.

We resected the medial meniscus *ligament* (not the medial meniscus) in all of our mice, as indicated in our Methods text: “**DMM surgery-induced OA was induced in C57BL/6J WT mice by surgically removing the medial meniscus ligament from the right knee joint of the hind limb as described previously^{18, 19, 62, 63}.**” This procedure follows that described by Glasson et al. in (Refs 18 and 63).

18. Glasson SS, Blanchet TJ, Morris EA. The surgical destabilization of the medial meniscus (DMM) model of osteoarthritis in the 129/SvEv mouse. *Osteoarthritis Cartilage* 15, 1061-1069 (2007).
63. Glasson SS, Chambers MG, Van Den Berg WB, Little CB. The OARSI histopathology initiative - recommendations for histological assessments of osteoarthritis in the mouse. *Osteoarthritis Cartilage* 18 Suppl 3, S17-23 (2010).

We agree that the OARSI score was unusually high for DMM surgery-induced OA. This reflected the use of a poorly standardized OARSI scoring system. We have addressed this as detailed in our response to Q1–3, namely, by repeating all animal OA/OA-like model experiments with an automated and very stringent OARSI scoring system (which is detailed in new Supplementary Table 4). Consequently, the average OARSI score after 8-week DMM now averages 2.9 points. Exact averages can be found in the accompanying Source Data. The new data are described in our responses to Q1–3.

To address the other points, we checked all tissues at sacrifice to confirm that destabilization of the medial meniscus had occurred.

We also deleted some text related to ligaments to avoid any potential confusion between the ACLT and DMM OA models.

Q18.

In Fig 5 one cannot use ANOVA for the OARSI score which is a semi-quantitative score. A non-parametric test should be used instead. In reporting the results of Fig. 6 the authors claim that “5-ASA treatment in OA protects against cartilage destruction by downregulating OSCAR and ECM-degrading enzymes” whereas a simple association is shown. I would suggest toning down the statement with 5-ASA treatment in OA protects against cartilage destruction and is associated with downregulation of OSCAR and ECM-degrading enzymes.

Reply: As indicated in our reply to Q9, we used Kruskal-Wallis test followed by Mann-Whitney *U* test instead of ANOVA for comparing groups in terms of OARSI scores.

We also modified the rather brash title of the section on Figure 6 according to your suggestion:

P18 L426-L427 “5-ASA treatment in OA protects the cartilage from destruction and associates with downregulation of OSCAR and ECM-degrading enzymes”

Q19.

Last year, Li et al. PMID: 36381242 reported the anabolic role of 5-ASA in chondrocytes and osteoarthritis, albeit they did not make the link with OSCAR and without the in vivo experimentation of this manuscript. Although this paper does not take away any of the novelty of this study, it still needs to be referenced and commented upon.

Reply: We strongly agree. While we did cite Li et al. in the previous manuscript (Reference #56), our text did not sufficiently recognize the importance of this paper for our study. To address this, we added the following texts to the Discussion:

P20-21 L455-462 “Here, we report our discovery that 5-ASA can serve as an OSCAR antagonist that effectively reduces DMM-induced OA and the cartilage destruction generated by OSCAR overexpression in the joint. It should be noted that during writing of this paper, Li et al.⁴⁸ reported that culturing human osteochondral explant models of OA with 5-ASA downregulated cartilage degradation. While they did not link this effect to OSCAR or conduct in vivo experimentation, these findings are highly consistent with our own, which together suggest that 5-ASA may have marked chondroprotective effects.”

P21 L472-476 “...we found that 5-ASA competed with collagen for binding to OSCAR and its binding to OSCAR upregulated PPAR γ . This downregulated the eicosanoid pathway, including COX-2 and its generation of PGE2 (Supplementary Fig. 8). The human osteochondral explant study of Li et al.⁴⁸ also observed that culture of the explants with 5-ASA downregulated COX-2.”

Minor points

Q20.

The results contain many aspects of methods and introduction. This is particularly evident, for instance, in end of p. 5 and the first half of p. 6.

Reply: We agree and have corrected this, as follows:

P6 L118-132 “**The extracellular OSCAR domain binds to collagen-II via the GPOGPAGFO consensus sequence. Moreover, this peptide (denoted as COL^{pep}) binds both to recombinant OSCAR protein and cell-surface OSCAR on chondrocytes¹⁴. It should be noted at this point that while normal chondrocytes express OSCAR-mRNA/protein at negligible levels, we found that this expression is upregulated *in vitro* by IL-1 β ¹⁴, which stimulates chondrocytes to produce cartilage-destroying enzymes such as MMPs and aggrecanase^{6,21}. Moreover, culture of normal chondrocytes on COL^{pep}-coated plates also increases their native OSCAR transcription, and IL-1 β augments it further (Supplementary Fig. 2a). While unstimulated Ad-OSCAR-infected chondrocytes express ~300-fold more OSCAR than IL-1 β +COL^{pep}-stimulated uninfected chondrocytes (Supplementary Fig. 2b), IL-1 β and/or COL^{pep} also augment it (Supplementary Fig. 2c). These OSCAR-transcription responses reflect downstream signaling induced by COL^{pep}-bound OSCAR that triggers new OSCAR expression¹⁵. These responses served as readouts of the binding of chondrocyte-surface OSCAR to COL^{pep} in our study.”**

Q21.

Purely as a suggestion, in patients with ulcerative colitis 5-ASA is given orally. While this is not required for this manuscript, I think that there is an important opportunity if it was also active systemically, because osteoarthritis, in clinical practice, usually involves multiple joints and because the drug is already approved for clinical purposes.

Reply: Indeed, 5-ASA is already clinically approved and is administered orally to patients with ulcerative colitis, and it would be of great interest to see whether such systemic administration would also be effective for OA. We are currently planning these studies. We added some text to the Discussion about this. Thank you very much for this point, and your extremely pertinent and helpful review in general.

P23 L512-514 “**Thus, IA delivery of 5-ASA may have considerable therapeutic potential in OA. This also raises the possibility that systemic administration of 5-ASA that is not formulated for bowel selectivity could be useful for OA that involves multiple joints.”**

Reviewer #3 Comments:

This is a potentially very interesting paper showing convincingly that OSCAR, a receptor first described as driving osteoclast maturation, can be blocked by the repurposed drug 5-ASA. Generally the data are of high quality with several in depth analyses to attempt to understand the mechanism of action, which appears to be through PPAR γ . Importantly, this identifies a novel class of anti-inflammatory that could be easily repurposed to test in human OA.

Reply: We are very grateful for your thoughtful and thorough advice and very helpful review. In our view, addressing your comments has greatly improved our manuscript.

Comments1:

These data fit very well with other publications showing that PPAR γ KO mice develop accelerated disease (cited by authors) and that retinoic acid signalling, acting in a PPAR γ -dependent manner, is able to suppress mechanically induced inflammatory signalling and modify disease in surgically induced murine OA (DOI: 10.1126/scitranslmed.abm4054). The latter paper is not cited by the authors but is important as it sheds some further light on their findings.

Reply: Indeed, Zhu et al. supports the notion that suppressing DMM-induced inflammatory signaling and OA depends on PPAR γ . Thank you very much for this highly pertinent reference. It is now Reference #33 and the following texts were added to the manuscript:

P11-12 L258-266 “**This revealed eight putative 5-ASA targets. Three have already been noted in the analyses above, namely, PTGS2/COX-2, RXRA, and PPAR γ . The other five predicted targets were ERBB2, TBXAS1, NR1I2, PPAR δ , and RARG (Fig. 2g): these were also in the Flip^{catabolic} and Flip^{anabolic} gene-sets (Supplementary Table 2) and two (PPAR δ and RARG) were in the 14 TFs that we found were most strongly altered by 5-ASA treatment (Fig. 2e). Moreover, Zhu et al.³³ showed that an RXR agonist can suppress OA and that this relies on activation of PPAR γ . In addition, like PTGS2/COX-2, TBXAS1 is also a key enzyme in the eicosanoid pathway (Supplementary Fig. 4a, b).**”

P22 L495-L499 “**Moreover, PPAR γ agonists protect animal models from OA, and these effects are mediated by their anti-inflammatory properties *in vitro* and *in vivo*⁵³. In addition, Zhu et al.³³ showed recently that a retinoic acid metabolism blocking agent can suppress DMM-induced inflammatory signaling and OA, and that this depends on activation of PPAR γ .**”

Comments2:

For instance, this paper demonstrates that retinoic acid is the limiting factor in PPAR γ -RXR anti-inflammatory actions. It is therefore perhaps highly relevant that RXRs are also regulated by OSCAR/5-ASA, and explains why, in Figure 4C, the addition of 5-ASA is required for PPAR γ agonists to have their full function. Furthermore, in Zhu et al, MAPK signalling was not affected by the anti-inflammatory effect of RXR-PPAR γ , suggesting that the anti-inflammatory effect was at the level of gene transcription, not signalling. This is consistent with Extended data Figure 5. Taken together the data suggest that OSCAR is enhancing PPAR γ action by affecting PPAR γ levels as well as promoting RXR-dependent function and this leads to suppression of inflammation by transcriptional regulation. I would strongly recommend that

the authors revise their manuscript taking into consideration these complementary findings.

Reply: We strongly agree with the notion that the OSCAR/5-ASA regulatory pathway regulates not only PPAR γ but also RXR (Retinoid X receptor). We are planning to investigate the link with RXR and RXRA and PPAR γ more closely in our next experiments. To address this comment, we included the following text:

P12 L260-266 “The other five predicted targets were ERBB2, TBXAS1, NR1I2, PPAR δ , and RARG (Fig. 2g): these were also in the Flip^{catabolic} and Flip^{anabolic} gene-sets (Supplementary Table 2) and two (PPAR δ and RARG) were in the 14 TFs that we found were most strongly altered by 5-ASA treatment (Fig. 2e). Moreover, Zhu et al.³³ showed that an RXR agonist can suppress OA and that this relies on activation of PPAR γ . In addition, like PTGS2/COX-2, TBXAS1 is also a key enzyme in the eicosanoid pathway (Supplementary Fig. 4a, b).”

We also added the RXRA data to Fig. 4g, Supplementary Fig. 4b, and Supplementary Fig. 5f (see figs at the end of this reply), and mentioned these data in the Results text:

P15-16 L355-359 “Notably, 5-ASA also increased chondrogenesis in a dose-dependent manner, as shown by greater Alcian-Blue staining (Fig. 4f). Moreover, this not only associated with decreased *Oscar* transcription, it also associated with increased expression of *Pparg*, the Flip^{anabolic} DEG *Rxra*, and the three markers cartilage-anabolism genes collagen-II/*Col2a1*, *Acan*, and *Sox9*²⁶ (Fig. 4g).”

P14 L325-329 “Moreover, overexpressing *Pparg* had the same effect as 5-ASA in terms of the Flip^{anabolic} DEGs (and known markers of cartilage anabolism^{7,8}) *Rxra*, *Col2a1*, *Acan*, and *Sox9* mRNA expression by the IL-1 β +COL^{pep}-treated chondrocytes: both treatments increased these mRNA levels (Supplementary Fig. 5f).”

Fig. 4g

Supplementary Fig. 4b

Supplementary Fig. 5e

I have some additional specific comments:

Q1.

The emphasis on COX2 is based on correlative studies, not mechanistic ones and I think this should be down-played in the manuscript. COX1, COX2 and COX1/2 KO mice show no protection in DMM induced OA (<https://doi.org/10.1002/art.33324>), nor do COX 2 antagonists modify human or murine disease (OA or RA). I think these likely represent biomarkers of an inflammatory pathway rather than effector molecules in OA pathogenesis.

Reply: This is a very pertinent point and one also brought up by another reviewer. However, the notion that COX-2 inhibitors do not modify OA has been challenged recently by several systematic reviews. One found that some COX-2 inhibitors (e.g. celecoxib) can in fact modify cartilage, bone and synovial disease in OA (Nakata et al. 2018). Another pointed to preclinical evidence that shows intra-articular administration of COX-2 inhibitors (i.e. not systemic administration, which is the normal route for COX-2 inhibitors) had DMOAD effects (Timur et al. 2020). We added some text to the Discussion about this, as follows. Nonetheless, it certainly remains possible that COX-2 is indeed only a biomarker of the eicosanoid pathway. Therefore, we also qualified the title of the paper and other texts:

Added text:

P21-22 L472-486 Discussion "...we found that 5-ASA competed with collagen for binding to OSCAR and its binding to OSCAR upregulated PPAR γ . This downregulated the

eicosanoid pathway, including COX-2 and its generation of PGE2 (Supplementary Fig. 8). The human osteochondral explant study of Li et al.⁴⁸ also observed that culture of the explants with 5-ASA downregulated COX-2. It should be noted that while it is widely believed that systemic administration of COX-2 inhibitors (i.e. common nonsteroidal anti-inflammatory drugs) does not associate with chondroprotective effects in OA, a recent systematic review suggested that some COX-2 inhibitors could act as DMOADs if they are injected IA⁵⁰. Moreover, another recent systematic review found that even when injected systemically, some COX-2 inhibitors (e.g. celecoxib) can modify the cartilage, bone, and synovial disease in OA, and these effects are mediated by the regulation of prostaglandins and direct changes to the tissues⁵¹. Since our 5-ASA treatment in mice involves IA injections and is potentially a novel drug for OA, it is thus possible that COX-2 and LOX-5 could be effectors of OA. Alternatively, they are simply markers of the eicosanoid pathway.”

Altered texts:

P1 L1-2 Paper title “5-aminosalicylic acid suppresses osteoarthritis through the OSCAR-PPAR γ axis”

P12 L268-269 “PPAR γ signaling in chondrocytes may mediate the cartilage-protective effects of 5-ASA”

P12 L282-286 “Since elucidating the molecular mechanisms by which 5-ASA protects cartilage could reveal therapeutic targets, we focused further on COX-2 and LOX-5 as either markers of the eicosanoid pathway or true targets. In either case, exploring the link between OSCAR, PPAR γ , COX-2, and LOX-5 could help illuminate the key role of PPAR γ in OA pathogenesis.”

The systematic reviews were also added as references:

P41 L1030-1036 References

50. Timur UT, *et al.* Chondroprotective Actions of Selective COX-2 Inhibitors In Vivo: A Systematic Review. *Int J Mol Sci* **21**, (2020).
51. Nakata K, *et al.* Disease-modifying effects of COX-2 selective inhibitors and non-selective NSAIDs in osteoarthritis: a systematic review. *Osteoarthritis Cartilage* **26**, 1263-1273 (2018).

Q2.

I would also down-play the distinct role that 5-ASA has on chondrogenesis/repair. Inflammatory activators (such as IL1, TNF etc) are known to promote degradation whilst also suppressing anabolism. The effect of 5-ASA may therefore be indirect through the reduction in inflammation. The effect sizes in these anabolism pathways are also modest.

Q4.

Do the authors have histology at 5 weeks i.e. prior to treatment with late 5-ASA (Fig 6). This would help establish whether there is reversal of disease or whether treatment simply halts progression from that time point.

Reply to Q2 & Q4: While conducting additional experiments for the current revision, we obtained additional evidence for the possibility that 5-ASA may promote chondrogenesis/repair. Specifically, we found that DMM mice that were injected with 5-ASA at weeks 8–11 and killed at week 12 had thicker cartilage than DMM mice that were injected with vehicle at weeks 4–7 and killed at week 8. This experiment, which was suggested by Reviewer 2 (Q16), together with our MSC experiments suggest that 5-ASA may have an anabolic effect in damaged cartilage. However, we recognize that additional experiments are needed to confirm this, including with in vivo models of cartilage regeneration. Therefore, we have been very careful to describe these data in a qualified manner. The new data are shown in Fig. 6. The following texts were added:

P2 L39-41 Abstract **“Moreover, mice with DMM-induced OA that were treated with 5-ASA at weeks 8-11 and sacrificed at week 12 had thicker cartilage than untreated mice that were sacrificed at week 8.”**

P4 L81-82 **“Late injections at weeks 8-11 also led to thicker cartilage compared to untreated mice that were sacrificed at week 8.”**

P18 L414-424 **“We then showed that 5-ASA could even induce DMM surgery-damaged cartilage to recover when it was first started after the disease had progressed to severe degenerative OA, namely, 8 weeks after DMM (Fig. 6a). Hence, mice subjected to a 4-week treatment with 5-ASA from weeks 8-11 exhibited a decrease in several OA markers when contrasted with mice at the 12-week post-DMM surgery stage. In fact, the final OARSI grade of the 5-ASA-treated mice sacrificed at week 12 was much lower than that of untreated mice that were sacrificed at week 8. This was also true for the other OA indicators. Significantly, the treated mice that were killed at week 12 had thicker cartilage than the mice that were sacrificed at week 8 (Fig. 6b, c). These findings suggest that 5-ASA could not just arrest OA progression, it could reverse it. However, additional studies are needed to confirm this.”**

P54 L1225-1234 Fig. 6 legend. **“a Mice were subjected to sham operation or DMM surgery and then IA-injected once per week for the indicated 4-week prior to sacrifice with vehicle (Veh; G1 and G2) or 5-ASA (2 mg/kg; G3) and sacrificed 9 (G1) or 12 weeks (G2 and G3) after the operation (n = 10 mice per group). b Representative Safranin-O staining and fast green counter staining images. Scale bar, 25 μm. c Quantitation of the following OA variables: OARSI grade, osteophyte maturity, subchondral bone plate thickness, and synovitis. The OARSI grade, synovitis and osteophyte maturity data are shown as means ± 95% confidence intervals (CI). Differences between groups were determined with Kruskal-Wallis test followed by Mann–Whitney U test. Means ± s.e.m. with two-tailed t-test for SBP thickness. Exact P values can be found in the accompanying Source Data. Scale bars, 25 μm.”**

Fig. 6b, c

Q3.

Where they are over-expressing OSCAR in the joint by Ad, I think it would be more accurate to describe this as OA-like disease rather than OA. It is an artificial model that does not mirror natural OA, and it looks as though there is principally proteoglycan depletion rather than breakdown of the cartilage (interestingly this is what is seen in the SOX9-aggrecan KO mice, (doi: 10.1002/jbmr.1696.)).

Reply: This was also a point raised by another reviewer, and we agree very much with it. Indeed, 3 weeks after weekly IA injections of Ad-OSCAR, the cartilage damage is limited to loss of proteoglycans and synovitis. However, when we extend the Ad-OSCAR injections for another 5 weeks (i.e. 8 weeks of treatment), we do observe changes in the OARSi grade, subchondral bone plate thickness, and osteophyte formation. So structural changes to the cartilage do eventually arise if the Ad-OSCAR injections are given long enough. We do agree, however, that we should be more cautious in our description of these changes. Therefore, we have made sure to refer to the joint changes in the Ad-OSCAR-injected mice as “OA-like” throughout the manuscript. We also made this point early in the Results:

P5 L97-106 “...we IA-injected murine knees with Ad-OSCAR (3 weekly injections; Fig. 1c). By the end of the third week, this induced *in vivo* chondrocyte-overexpression of OSCAR (Fig. 1d), damaged the glycosaminoglycans in the articular cartilage, and induced synovitis. Notably, there was no obvious cartilage loss or osteophyte development, and the thickening of the subchondral bone plate (SBP) that is suggestive of sclerosis was not observed (Fig. 1e-f). However, as will be detailed later, when mice were treated for 8 weeks with IA Ad-OSCAR injections, cartilage destruction, osteophyte formation, SBP thickening, and synovitis were observed (Fig. 1h-i). Nonetheless, it should be noted that the Ad-OSCAR-induced model involves OA-like disease rather than classical OA.”

Q5.

Could the authors provide the list of the 124 first hit compounds in their screen and which ones have already been studied in OA – this would be of interest to the readers.

Reply: The 124 compounds that initially inhibited the binding of OSCAR to collagen in our screening study are listed in Supplementary Table 1. For clarification, 124 of the 165 compounds were excluded because their roles in chondrocytes were already known or they had been studied in bone disease. We have revised the manuscript to clarify this:

P7 L140-142 “**Of the 165 primary target compounds that emerged, 124 compounds were excluded because their role in chondrocytes or OA has already been studied (Supplementary Fig. 2f; Supplementary Table 1).**”

Q6.

There are several important discussion points that I think are missing.

Firstly, the authors say very little about what they think is happening in bone. Are the same inflammatory pathways activated??

Reply: This is an exceedingly interesting question because OSCAR was first shown to be a highly expressed receptor on osteoclasts (Kim N. et al.; ref #9) and is thus widely known for its role in bone. In fact, this role is so well-established that OSCAR is used as a marker of bone resorption. Our research team, with a well-established history of concentrating on bone homeostasis, was the initial proponent of considering OSCAR as a prospective therapeutic target for cartilage, diverging from its customary affiliation with bone resorption (Kim N et al.; ref 9). This led to our discovery that OSCAR is upregulated on chondrocytes in OA (Park DR et al.; ref 14) and, in the present study, that 5-ASA can inhibit OSCAR in chondrocytes. Our *in vivo* and *in vitro* experiments in bone, which we conducted alongside our cartilage experiments, have now also shown that 5-ASA either inhibits OSCAR in bone and/or regulates bone remodeling independently of OSCAR. We plan to publish these studies in a follow-up paper after this manuscript is published. In order to preserve the novelty for future investigations, we have intentionally provided limited information in this paper regarding the indications that 5-ASA's capacity to regulate OSCAR extends beyond chondrocytes and encompasses osteoclasts. We anticipate that these aspects will be comprehensively explored in forthcoming studies.

P38 L887-888; P39 L904-905 References

9. Kim N, Takami M, Rho J, Josien R, Choi Y. A novel member of the leukocyte receptor complex regulates osteoclast differentiation. *J Exp Med* **195**, 201-209 (2002).
14. Park DR, *et al.* Osteoclast-associated receptor blockade prevents articular cartilage destruction via chondrocyte apoptosis regulation. *Nat Commun* **11**, 4343 (2020).

Q7.

Is this collagen II peptide activated (presume not). Is the collagen peptide exclusive to type II collagen or is it also in other collagens (type I and VI for instance)? And is it exposed on the native collagen or only exposed after degradation?

Reply: Barrow *et al.* (ref #10) provides evidence of OSCAR's binding affinity to collagens I, II, and III under *in vitro* conditions. Notably, the surface of natural bone is enveloped by a layer of fibrillar collagen, typically concealed beneath osteoblastic bone-lining cells. The investigation delves into the question of whether these collagens are accessible to mononuclear osteoclast precursors expressing OSCAR in a live, *in vivo* context. The findings indicate that collagens I and III are discernible on active bone surfaces, contrary to quiescent ones, and are in contact with OSCAR-expressing cells within human bone biopsies. Importantly, *in vivo* experiments have unveiled that this peptide's exposure occurs exclusively when native collagen sustains damage. In such instances, the peptide assumes the role of a damage-associated molecular pattern (DAMP). These critical insights have been elucidated in the revised manuscript.

P38 L890-891 References

10. Barrow AD, *et al.* OSCAR is a collagen receptor that costimulates osteoclastogenesis in DAP12-deficient humans and mice. *J Clin Invest* **121**, 3505-3516 (2011).

P20 L436-443 **“The OA hallmark is tipping of the fine balance between chondrocyte production of ECM and chondrocyte-mediated ECM catabolism: the latter predominates, which results in cartilage destruction. This reflects mechanical stress⁴⁶, which generates ECM fragments (including COL^{pep}) that are recognized as damage-associated molecular patterns (DAMPs) by receptors on chondrocytes. The DAMP signals also induce proinflammatory cytokines that promote ECM catabolism and interrupt ECM anabolism. These factors together decrease chondrocyte numbers, which reduces their ability to maintain cartilage homeostasis⁴⁷.”**

Q8.

5-ASA is given to patients with UC orally with bioavailability designed to be bowel selective. Are oral preparations available that have more systemic effects?

Reply: We found this question very interesting. It was also raised by another reviewer. Since drugs to treat knee osteoarthritis are typically administered intra-articularly, we used this route to administer 5-ASA in our paper. However, the fact that 5-ASA is administered orally in

inflammatory bowel disease suggests that oral 5-ASA might also be useful for multi-joint osteoarthritis. To our knowledge, oral preparations of 5-ASA that are not selective for the bowel are not yet available but we intend to explore this in future studies. To address this point, we added the following text to the Discussion:

P23 L512-514 “Thus, IA delivery of 5-ASA may have considerable therapeutic potential in OA. This also raises the possibility that systemic administration of 5-ASA that is not formulated for bowel selectivity could be useful for OA that involves multiple joints”

Q9.

Why does overexpression of OSCAR in vivo lead to OA-like disease in the absence of col II peptide trigger? Does this mean that OSCAR can signal ligand-independently when overexpressed?

Reply: This is also an excellent question to which we do not yet have an answer. We do not know how OSCAR overexpression induces OA-like disease, although we do not believe that these OSCAR molecules can transmit ligand-independent signals. It is possible that the IA injections induce the release of some COL^{pp}. Another possibility relates to the fact that OSCAR, which lacks a cytoplasmic tail, acts together with FcR γ as a co-receptor for signaling transduction that leads to osteoclastogenesis (Merck E et al.; ref 13, mentioned in P3 L64-66): it is possible that overexpressing OSCAR activates its co-receptor, which then triggers pathogenic OA mechanisms. We intend to test these hypotheses in our future research. To address this point, we added the following text to the end of the Discussion:

P23-24 L529-531 “Research on the mechanisms by which OSCAR overexpression alone drives OA pathogenesis is also warranted.”

P38 L900-902 References

13. Merck E, *et al.* OSCAR is an FcR γ -associated receptor that is expressed by myeloid cells and is involved in antigen presentation and activation of human dendritic cells. *Blood* **104**, 1386-1395 (2004).

Minor comments:

Q10.

There is a lot of result in the latter part of the introduction which could be removed. Similarly there is quite a lot of discussion in the results e.g. lines 217-231.

Reply: During revision, we redistributed the Introduction, Results, and Discussion sections to improve the presentation. As a result, we removed the Results section at the end of the Introduction and reduced the unnecessary discussion in the Results.

Q11.

Extended Figure 2 has an extra panel (k) with consequent shunting of the other figure panels – the text is therefore out of synch.

Reply: Thank you very much for picking this error up. During revision, we ensured that all display items are correctly labeled and cited.

Q12.

Line 113 please avoid saying “IL1b ...plays key pathogenic roles in OA” (e.g in lines 112 and 331, 340) - there is very little direct evidence to support a role for IL1 in knee OA either in mice (in KO studies) or in human (therapeutic studies) despite many people trying to demonstrate this. Many other cytokines or stimuli (e.g. mechanical damage directly) activate the same pathways as IL1b.

Reply: We agree and have replaced all language that suggests a direct link between IL-1 β and OA with language that describes the increased expression of MMPs by chondrocytes as being due to treatment with IL-1 β . The following correction has been made:

P6 L120-124 “It should be noted at this point that while normal chondrocytes express OSCAR-mRNA/protein at negligible levels, we found that this expression is upregulated *in vitro* by IL-1 β ¹⁴, which stimulates chondrocytes to produce cartilage-destroying enzymes such as MMPs and aggrecanase^{6,21}.”

Q13.

Also, there is no direct evidence that MMP3 is pathogenic. Indeed, MMP3-/- mice are not protected from DMM induced OA (several groups have shown this). Suggest remove the two references to MMP3.

Reply: We removed the two references to MMP3 and replaced them with more appropriate references (ref #6). We also ensured that it is clear that MMP3 is upregulated in response to inflammatory cytokines such as IL-1 β and that MMP3 expression is not definitive evidence of pathogenicity.

P3 L58-62 “In OA, this balance is tipped towards catabolism mediated by the matrix-metalloproteinases MMP3, MMP9, MMP13, and the aggrecanase ADAMTS^{5,6}. These degrade collagen-II and aggrecan (ACAN), whose synthesis is also downregulated in OA^{7,8}. Thus, targeting chondrocyte catabolic/anabolic processes, thereby modifying cartilage-ECM structures, may yield promising OA therapies.”

6. Wang J, Markova D, Anderson DG, Zheng Z, Shapiro IM, Risbud MV. TNF- α and IL-1 β promote a disintegrin-like and metalloprotease with thrombospondin type I motif-5-mediated aggrecan degradation through syndecan-4 in intervertebral disc. *J Biol Chem* **286**, 39738-39749 (2011).

Q14.

In line 246-7, is this result referring to Figure 1f? In which case, please add this. Line 110 please add reference to the collagen peptide binding to OSCAR.

Reply: We reorganized the description of line 246-7 in the revised version.

In previous manuscript, L246-247 “Moreover, the PPAR γ , RXRA, and RARG FlipTFs appeared to associate particularly strongly with Flip^{catabolic} DEGs.”

p11-12 L255-263 “**This analysis indicated how closely the 5-ASA-induced Flip-DEG profile resembled the DEG profiles induced by each of the >20K drugs/compounds. We then took the top 10% most similar CMap profiles and listed the most enriched targets (see Methods for details). This revealed eight putative 5-ASA targets. Three have already been noted in the analyses above, namely, PTGS2/COX-2, RXRA, and PPARG. The other five predicted targets were ERBB2, TBXAS1, NR1I2, PPARD, and RARG (Fig. 2g): these were also in the Flip^{catabolic} and Flip^{anabolic} gene-sets (Supplementary Table 2) and two (PPARD and RARG) were in the 14 TFs that we found were most strongly altered by 5-ASA treatment (Fig. 2e).**”

We also added references regarding the binding of collagen peptide to OSCAR, as follows.

P6 L118-120 “**The extracellular OSCAR domain binds to collagen-II *via* the GPOGPAGFO consensus sequence¹⁰. Moreover, this peptide (denoted as COL^{pep}) binds both to recombinant OSCAR protein and cell-surface OSCAR on chondrocytes¹⁴.**”

REVIEWERS' COMMENTS

Reviewer #1 (Remarks to the Author):

The authors adequately addressed my comments from the previous round of review.

Reviewer #2 (Remarks to the Author):

The authors have addressed all my queries and I would like to congratulate them for their impressive, and very exciting manuscript.

Having repeated all the joint destabilization experiments and re-scored, now the structural damage and also the bone sclerosis is well evident even in the early time points. Therefore the authors may want to be less cautious and just claim osteoarthritis in their in-vivo experiments.

Another suggestion: whenever there is more than one group, usually a generalised linear model followed by comparison of the estimated marginal means is more powerful than a Kruscall-Wallis test and clearly also allows a more sophisticated analysis, including the interaction of multiple independent variables. Not strictly required here.

Reviewer #3 (Remarks to the Author):

The authors have addressed all of my comments and have added significant new data, including robust demonstration that 5-ASA is able to reverse established disease in vivo.

I have two minor comments regarding the discussion. Firstly, their new opening paragraph indicates that mechanical stress induces OA by release of matrix fragments. I think it is important that they qualify this by saying that one mechanism is by release of matrix fragments. There are multiple ways in which chondrocytes respond to mechanical stress which are independent of matrix fragments. The second minor comment relates to their reference of the use of PPAR γ agonists in murine models in vivo (ref 53),

line 497. They cite a review here rather than the primary source. I am aware that others have failed to show disease modification with PPAR γ agonists (unpublished) and so this citation, if robust, is important.

In response to Reviewers' comments:

Reviewer #1 (Remarks to the Author):

The authors adequately addressed my comments from the previous round of review.

Reply: We appreciate your positive review.

Reviewer #2 (Remarks to the Author):

The authors have addressed all my queries and I would like to congratulate them for their impressive, and very exciting manuscript.

Having repeated all the joint destabilization experiments and re-scored, now the structural damage and also the bone sclerosis is well evident even in the early time points. Therefore the authors may want to be less cautious and just claim osteoarthritis in their in-vivo experiments.

Another suggestion: whenever there is more than one group, usually a generalised linear model followed by comparison of the estimated marginal means is more powerful than a Kruskal-Wallis test and clearly also allows a more sophisticated analysis, including the interaction of multiple independent variables. Not strictly required here.

Reply: We appreciate your positive review. In our upcoming analyses of multiple groups, we plan to adopt a more advanced method, specifically using a generalized linear model and subsequently comparing the estimated marginal means, as a preferred alternative to the Kruskal-Wallis test. We would like to express our sincere gratitude once again for your valuable guidance and suggestions.

Reviewer #3 (Remarks to the Author):

The authors have addressed all of my comments and have added significant new data, including robust demonstration that 5-ASA is able to reverse established disease in vivo.

I have two minor comments regarding the discussion.

Firstly, their new opening paragraph indicates that mechanical stress induces OA by release of matrix fragments. I think it is important that they qualify this by saying that one mechanism is by release of matrix fragments. There are multiple ways in which chondrocytes respond to mechanical stress which are independent of matrix fragments. The second minor comment relates to their reference of the use of PPAR γ agonists in murine models in vivo (ref 53), line 497. They cite a review here rather than the primary source. I am aware that others have failed to show disease modification with PPAR γ agonists (unpublished) and so this citation, if robust, is important.

Reply: Thank you for your insightful comments. We have carefully considered each point and are pleased to provide a detailed response outlining the revisions we have made in accordance.

Q1.

Their new opening paragraph indicates that mechanical stress induces OA by release of matrix fragments. I think it is important that they qualify this by saying that one mechanism is by release of matrix fragments. There are multiple ways in which chondrocytes respond to mechanical stress which are independent of matrix fragments.

Reply: We agree and have revised the new opening paragraph to clarify that mechanical stress is not the sole factor contributing to the degradation of the extracellular matrix (ECM) into matrix fragments. The revised paragraph is as follows:

P19 L433-438 **“The OA hallmark is tipping of the fine balance between chondrocyte production of ECM and chondrocyte-mediated ECM catabolism: the latter predominates, which results in cartilage destruction. This reflects that enzymatic degradation, oxidative stress, and mechanical stress⁴⁶ lead to the production of ECM fragments (including COL^{pep}), which are recognized as damage-associated molecular patterns (DAMPs) by receptors on chondrocytes.”**

Q2.

The second minor comment relates to their reference of the use of PPAR γ agonists in murine models in vivo (ref 53), line 497. They cite a review here rather than the primary source. I am aware that others have failed to show disease modification with PPAR γ agonists (unpublished) and so this citation, if robust, is important.

Reply: We very much agree and greatly appreciate your suggestion. We therefore have substituted the existing ref 53 with a primary source citation from Kobayashi et al.

P40 L1035-1037 References

53. Kobayashi T, et al. Pioglitazone, a peroxisome proliferator-activated receptor gamma agonist, reduces the progression of experimental osteoarthritis in guinea pigs. *Arthritis Rheum* 52, 479-487 (2005).